# Fostering Innovation, Transition, and the Reconstruction of Forestry: Critical Thinking and Transdisciplinarity in Forest Education with Strategy Games

Patrick O. Waeber [1,2,*], Mariana Melnykovych [1], Emilio Riegel [3], Leonel V. Chongong [3], Regie Lloren [3,4], Johannes Raher [1], Tom Reibert [3], Muhammad Zaheen [3], Oleksandr Soshenskyi [5] and Claude A. Garcia [1,2,*]

[1] Forest Policy and International Forest Management, Department of Agricultural, Forest and Food Sciences HAFL, Bern University of Applied Sciences, Länggasse 85, 5032 Zollikofen, Switzerland; mariana.melnykovych@bfh.ch (M.M.); johannes.raher@students.bfh.ch (J.R.)

[2] Ecosystem Management, Department of Environmental Systems Science, Swiss Federal Institute of Technology (ETH), Universitätsstr. 16, 8092 Zürich, Switzerland

[3] Department of Forestry, Weihenstephan-Triesdorf University of Applied Sciences, Hans-Carl-von-Carlowitz-Platz 3, 85354 Freising, Germany; emilio.riegel@student.hswt.de (E.R.); leonel.chongong@student.hswt.de (L.V.C.); regie.lloren@student.hswt.de (R.L.); tom.reibert@student.hswt.de (T.R.); muhammad.zaheen@student.hswt.de (M.Z.)

[4] Institute of Agriculture, Camiguin Polytechnic State College, Tangaro, Catarman 9104, Philippines

[5] Institute of Forestry and Landscape-Park Management, National University of Life and Environmental Sciences of Ukraine, Horikhuvatskyi Shliakh 19, 03041 Kyiv, Ukraine; soshenskyi@nubip.edu.ua

* Correspondence: patrick.waeber@bfh.ch (P.O.W.); claude.garcia@bfh.ch (C.A.G.)

**Abstract:** Forest education plays a crucial role in achieving the Sustainable Development Goals and promoting sustainable forest management amidst global challenges. However, existing programs struggle to keep pace with rapidly evolving crises and uncertainties that contribute to deforestation and forest degradation. To tackle these challenges, integrating innovative approaches into forest education is essential. This paper showcases the transformative use of a strategy game, MineSet, as an innovative teaching method for integrated forest management. The game facilitates deeply engaging experiences that provide unique insights into complex issues like deforestation. By assuming various stakeholder roles, graduate students actively engage with and confront the intricate tradeoffs inherent in forest management. This interactive and immersive role-play game not only fosters critical thinking skills but also promotes collaborative problem-solving, making MineSet a highly innovative and attractive tool in forest education. The importance of extended debriefings, facilitation throughout the game, and ongoing discussions should not be underestimated, as they establish meaningful and necessary connections between in-game events, validated educational material, and published research outcomes. Moreover, the game equips students with practical experience and a comprehensive understanding of landscape approaches, using the Congo Basin as a case study. We emphasize the potential of innovative forest education to foster sustainability, stimulate critical thinking, resolve conflicts, and prevent costly forest losses.

**Keywords:** MineSet; innovations in teaching; transformative learning; serious games; RPGs; role-play games; sustainable forest landscape governance; social innovation

## 1. Introduction

The importance of forests in achieving the United Nations Sustainable Development Goals (SDGs) is widely recognized [1–4]. However, the recent literature suggests that forest education is not meeting the rapidly changing demands of the labor market, and the significance of forests and forest-related professions is often undervalued [5,6]. To ensure that forests make a maximum contribution to the SDGs, a workforce proficient in forestry

and related disciplines, as well as widespread public knowledge of forest-related issues based on broad ranges of disciplines and knowledge systems, is essential [7].

Forest education plays a vital role in developing the necessary knowledge, skills, and values for sustainable forest management and achieving environmental, social, and economic goals at various levels [6,8–10]. However, scholarly sources highlight the shortcomings in forest-related education, including inadequacy, outdatedness, and insufficiency, leading to limited awareness and understanding of forests and poorly equipped forest graduates who are ill-prepared for the changing workplace demands [11]. Challenges persist globally, such as low student enrolment in professional forest education programs and limited integration of forest-related subjects in curricula [12].

The forest-based sector has experienced changes in recent years due to evolving societal demands, leading to new trends in forest education globally [13,14]. These changes are reflected in the labor market and the expectations of students for more diverse experiences and skills. The need for real-life solutions to problems such as climate change calls for holistic and cross-sectoral approaches, which have led to changes in the university curricula towards more multidisciplinary programs. However, despite an overall increase in the number of forestry graduates and advancements towards gender parity, forest education has been insufficiently addressed in existing international efforts and the collaborative efforts between leading institutions that have started to emerge. Forest education has been missing from the global forest policy agenda for nearly 20 years [15]. However, there has been renewed interest in forest education, as reflected in the increased activities of research organizations and non-governmental organizations (NGOs) and its inclusion on the agenda of the 14th session of the United Nations Forum on Forests [16,17]. The International Union of Forest Research Organizations (IUFRO) and the International Forestry Students Association (IFSA) Joint Task Force on Forest Education as a collaborative project aims to bring together NGOs, researchers, and students to shape the future of forest education, strengthen forest education, and highlight ways to make the sector attractive to young people [18]. This signals a growing realization that forest education can be part of the solution to reducing deforestation and forest degradation, protecting ecosystems, enhancing livelihoods and human health, conserving biodiversity, and mitigating and adapting to climate change [11,19]. Strengthening forest education is essential for sustainable forest management and achieving global development goals [12].

Forest education of graduate students must adapt to the numerous challenges facing the forest sector, which includes changes in societal expectations regarding forest goods and services, alterations in employment trends, and a lack of interest in the sector [20]. Furthermore, the aging workforce in many countries and outdated curriculums must be addressed. Urgently, forest education needs to be revitalized and expanded, with new opportunities emerging from modern digital information and communication technologies and green economy jobs. Education systems must also integrate indigenous and traditional forest-related knowledge to manage and safeguard natural resources [11]. The Sustainable Development Goal 4, Quality Education, underlines the need for improved education on sustainable development. Without a resurgence in forest education, it will be difficult to achieve sustainable forest management and recognize the full value of forest goods and services. Additionally, without robust and suitable forest education, it is unlikely that forests and trees will fulfil their potential contributions to the global development goals, including the Sustainable Development Goals (SDGs), the targets of the United Nations Framework Convention on Climate Change (UNFCCC), the post-2020 Global Biodiversity Framework of the Convention on Biological Diversity (CBD), the United Nations Strategic Plan for Forests, and other global goals.

Considering the rapid pace of climate change and the threat of crossing planetary limits, the upcoming decade is crucial for the effective management of natural resources and environmental governance [21,22]. The issue of deforestation, loss of biodiversity, and increasing social disparities only heighten the urgency. Creating a space for scenario planning and exploring various paths is a vital tool in shaping environmental policy [23].

The objective of this article is to conduct a comprehensive examination of a strategy game that has been developed through participatory stakeholder engagement processes [24,25]. Our focus is on documenting and exploring the game's potential as an educational tool within the classroom setting. By immersing graduate students in a virtual reality of forest management, the game effectively guides them through evolving challenges, facilitating the development of decision-making skills relevant to real-world forest management scenarios. We provide a detailed account of the game's development, application, practical implementation, and the potential benefits it offers for forest education.

To achieve this objective, our article follows a structured approach. We begin with a brief literature review on the utilization of games in education. Subsequently, we present the game model in the Methodology section, along with its implementation in the classroom. The Results section provides a descriptive account of the gameplay and how the game's outcomes have been utilized for teaching purposes. Finally, we critically discuss the advantages and disadvantages of employing such a game in the classroom, focusing on its capacity to foster transitions, critical thinking, and transdisciplinarity in forestry education.

## 2. Literature Review

This section offers an overview of games within an educational context, with a focus on role-play games. Particularly, we introduce strategy games that have been collaboratively developed by stakeholders and scientists through a participatory process known as Companion Modelling [26,27]. These innovative games are utilized in the classroom, marking a novel approach in our educational practices.

Games are typically enjoyed for their entertainment value and possess the capacity to evoke a range of emotions, including happiness, during the gaming experience. On the other hand, serious or applied games are intentionally developed and utilized to fulfill a specific objective rather than solely for amusement. Serious games, a term coined by Abt [28], are specifically designed with an educational purpose [29]. Serious games serve as engaging educational tools that enable players to enhance their knowledge and skills by overcoming various challenges during gameplay. Throughout the gaming process, the performance of players is evaluated and scored [30,31]. Successful completion of obstacles leads to rewards, including scores, progression, and increased capabilities. Consequently, performance assessment plays a crucial role in serious games designed for knowledge acquisition and skill development. The game's system must accurately evaluate learner progress, as rewards and advancements depend on this assessment. Additionally, providing consequential feedback to players is essential for their continuous growth and improvement [30]. Moreover, educational elements can seamlessly integrate into the gameplay, allowing players to subconsciously acquire knowledge while fully immersed in the gaming experience [32,33].

Serious games have been widely recognized for their potential to elicit positive effects on learning motivation and outcomes [34–39]. Iten and Petko [40] found that while enjoyment was positively correlated with engagement, it had a limited impact on self-assessed or tested learning outcomes. They argue that explicit learning tasks and instructional support from the game or teachers are more influential in determining learning gains. This underscores the vital importance of instructional design, scaffolding, and educator guidance in facilitating and promoting meaningful learning experiences [41]. It further accentuates the necessity for a comprehensive approach that extends beyond mere enjoyment when employing serious games for educational purposes [40].

RPGs (role-playing games) are a specific type of serious game in which players take on different identities within the game. Role-playing games (RPGs) encompass various dimensions—rules, processes, spatial and temporal resolutions, characters, inter alia—and are generally classified into four types: Tabletop role-playing games, live-action role-playing games, computer role-playing games, and multiplayer online role-playing games [42] (but see also [43]). RPGs in their various forms have found meaningful applications in the field of education, highlighting their unique role compared to traditional educational role

play [44]—for a comparative analysis between RPGs and role play in education see [44,45]. The implementation of RPGs in an educational context encompasses a wide range of activities, e.g., arts-based research as an engaging and valuable medium for collaborative art-making [46], facilitating vocabulary learning strategies and mastery [47,48], promoting literacy skills and engagement [49], cultivating a sense of shared responsibility and encouraging the practice of making tradeoffs within the context of a common-pool management and conservation challenge [50]. RPGs strive to cultivate participants' adaptability, communication skills, leadership abilities, and a profound grasp of reality [43,51–54]. The integration of RPGs in education offers the opportunity to simulate group dynamics and present unique scenarios that foster creativity and critical thinking (e.g., encouraging players to revisit their assumptions and make adjustments and modifications to their existing beliefs as a result) [55]. RPGs have shown benefits in facilitating learning across subjects, enhancing literacy motivation, and improving social skills [56–60].

In this study, we employed a customized role-play game called MineSet, which specifically addresses the dynamics of central African forests and aims to educate participants about the underlying causes of deforestation and other global challenges [24,25]. Developed through a participatory modeling approach known as Companion Modeling [26], this constructivist-based game (cf. [61]) is not an off-the-shelf game but rather tailored and created to address a specific and intricate resource management issue in the Congo Basin. The MineSet game development, verification, and validation process included the active participation of various stakeholders from the region. Their involvement ensured that the game's assumptions were grounded and the relevant actors and resources and interactions properly calibrated.

Role-playing games, from haptic tabletop games, over computer-supported RPGs, to fully computerized Agent-Based Models, have a rich history of use in research and practice. These types of games have been employed and refined for over four decades in the fields of adaptive environmental management science and participatory action research [62–66]. Role-playing games have been proven to be effective tools for facilitating decentralized problem-solving and promoting a bottom-up approach that empowers participants to actively engage with complexity and uncertainty—in contrast, top-down prescriptive approaches that rely on expert-driven solutions have been shown to have limited effectiveness in managing complexity and uncertainty [67].

Strategy games can depict validated realities, thus generating a profound sense of realism for both players and beta testers [23]. When fully immersed in the game's virtual world, individuals adopt specific roles and perceive their experience as genuinely authentic [23]. We assume that by providing students with an immersive experience that allows them to actively engage and "be inside the system" of study, role-play games, particularly those co-developed by stakeholders, become a compelling addition as an educational tool in classrooms.

## 3. Materials and Methods

The game session described in this paper involved 6 students from the "International Management of Forest Industries" Master's program at the Bern University of Applied Sciences. This session took place in January 2023 and spanned one week, with continuous gameplay and debriefing sessions lasting 4 to 6 h per day. The participating students were in their final year of the Master's program and already had a strong background in forestry. The entire course, including the gameplay and discussions, was facilitated by the teachers. As the course was conducted online with students located off campus, we relied on an online whiteboard (Mural) to aid our discussions (which took place on Zoom) and to monitor our progress. Additionally, we utilized a role-playing game as our primary method for engaging in-depth discussions on forest change and deforestation drivers. Although traditionally played in person, we created a Mural-compatible beta version of the game.

The game allows for 5 to 14 players, with players assuming the role of a Chief Executive Officer (CEO) representing a logging or mining company. There are a total of 5

to 7 companies available, and if multiple players choose the same company, they act as a single CEO. Additionally, logging companies have the option to venture into mining operations. The logging aspect of the game involves interactions with the Ministry of Forests, while mining activities require permits and rights obtained from the Ministry of Mining. A company can hold more than one concession. CEOs must interact with markets, government, and non-governmental organizations (NGOs) to design strategies and shape the environment, economy, and society with their actions. The game considers the economic and financial constraints, the demographic development, governance, transparency, technical innovation, and cultural differences to reflect the major drivers of land use change in the tropics. The objective is for players to balance development and the conservation demands of the system. It has been used for academic research and workshops with a focus on forest management, logging, and mining activities [24,25]. The game can be played in solitaire, but in that case, the richness of interactions is reduced as all other roles need to be impersonated by the team of teachers.

During a game of MineSet, players gain a basic understanding of the intricate dynamics involved in land use change within the tropical forests of Central Africa. Through a step-by-step process, players are exposed to the complexities of the system, allowing them to deepen their knowledge and insight into the environmental, social, and economic factors in a playful manner. The game has loosely defined temporal and geographical scales. One hexagon is said to represent 500 km$^2$ and one turn 10 years, but none of these values have an impact on the flow of the game. They just help participants enter the game by providing them with familiar elements. To help players learn the rules by playing, the story begins in the 1960s with high-density forests on either side of a single road connecting two urban centers and minimal regulations in place. Farmers and hunter–gatherers live along this road—representing the resettlement imposed by colonial powers in the late 19th century to early 20th century. Each player is responsible for managing a logging or mining company with the same starting conditions. In total there are 9 concessions, which fall under the jurisdiction of the Ministry of Forestry; mining concessions are overlapping and fall under the jurisdiction of the Ministry of Mining. The concessions are located north and south of the already existing road connecting the east and west of the "gamescape" (the game landscape board). Throughout the game, players must compete to acquire logging concessions (during public auctions), construct roads, harvest timber (precious red wood or common brown wood), and sell their products on the global market.

In the game, logging concessions can be allocated to companies through leasing agreements. These leasing agreements determine which companies have the rights to operate in specific areas. However, there are also concessions that are not leased to any company. In the gameplay dynamics, players have the opportunity to engage in discussions and negotiations with the Ministry of Forestry regarding the transformation of certain concessions into protected areas. This introduces concepts such as land sparing, leakage, and strict conservation into the game. Each concession in the game has unique characteristics such as resources, biodiversity, exposure, and market access, leading to a distinct situation for each player. Concessions are randomly assigned to players at the beginning of the game, which comprises multiple phases. During the initial phase, players have a brief period to strategize and discuss amongst each other—potential collaboration, infrastructure development, access rights, shared investments, inter alia—before harvesting timber from their concession and sell it to the market. The decisions made by the players affect the landscape and provide insights into the impact of their choices. For instance, if a CEO chooses to allow logging operations by constructing roads through previously undisturbed forests, one of the outcomes is the arrival of migrants, represented as tokens in the game. This consequence becomes apparent to the players during gameplay, introducing an element of surprise and unpredictability. In the final phase of the round, players can choose to invest in improving their creditworthiness for future rounds or, alternatively, invest in, e.g., a sawmill or infrastructure or acquire additional concessions. There are several strategies to

play MineSet, depending on the goal of the player. Every strategy evolves and may change during the game session various times.

Additionally, players' strategic decisions are influenced by global events and expectations. During the 1960s and 1970s, players can prioritize road network development and schedule timber extraction, including precious red wood and common brown wood. During the 1980s, the concept of sustainability gained recognition, leading to an increasing significance of public opinions. Consequently, companies were compelled to adopt more environmentally friendly practices. As public opinion rose, credit ratings improved, resulting in greater access to loans from banks for increased investments. This allowed companies to expand their logging operations, highlighting the financial aspect of sustainability during that decade. In the 1990s, the focus shifted to environmental aspects, introducing biodiversity through an NGO promoting environmental studies and the hiring of biologists to evaluate biological diversity. Certification schemes like FSC offer access to regulated European markets or less regulated alternatives like Asian markets. The 2000s and 2010s highlight social justice, with advocacy groups playing a central role. Throughout the game, tensions escalate due to increased migration influenced by neighboring conflicts, impacting concessions and the wider landscape.

MineSet features all the major underlying drivers of land use change in the tropics, including demographics, economical and finance signals, governance and transparency, technological changes, and cultural differences (Figures 1 and 2). As the game unfolds, players discover the complexity of the system and devise new rules and strategies to balance development and conservation.

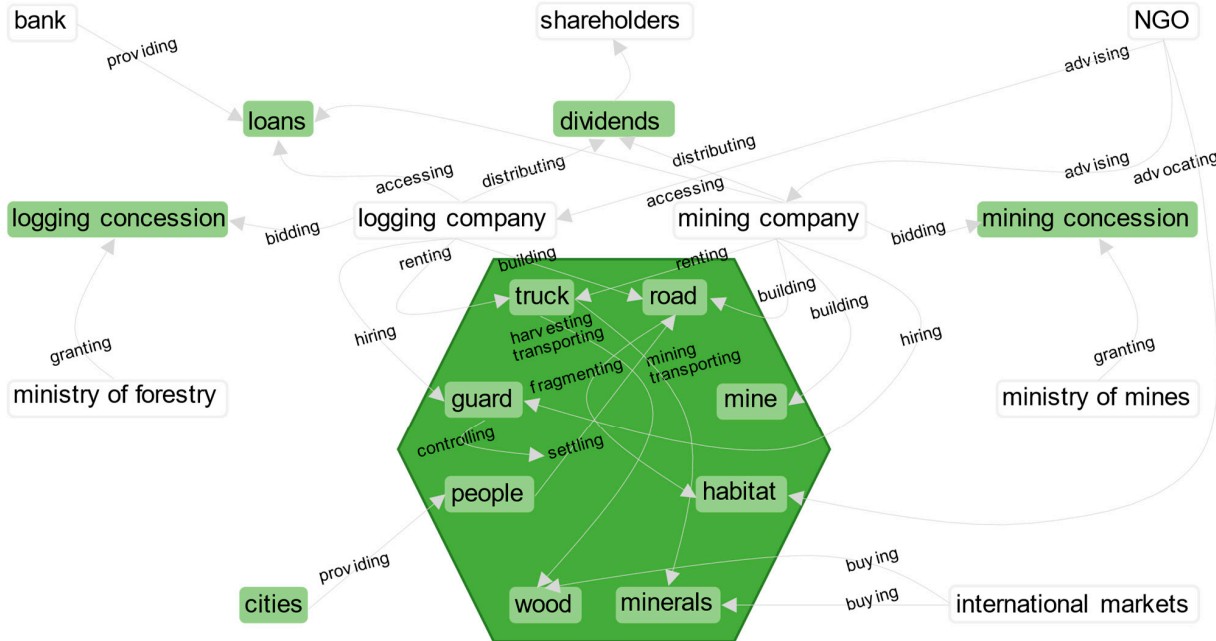

**Figure 1.** MineSet Conceptual Model. The model was developed through workshops for the Cofortips project in the Congo Basin [24,25]. It employed a participatory stakeholder engagement process using the ARDI (actors, resources, dynamics, and interactions) approach [68]. Actors are depicted by white boxes, resources by green boxes, and arrows symbolize the stakeholder–resource interactions. This conceptual model served as a blueprint for gamifying MineSet [24,25]. In the game, the "gamescape" consists of hexagons representing forest cover. Players assume roles like mining and logging companies, while non-playing characters are managed by the facilitation team (e.g., international markets, NGOs, bank, and ministries). The green hexagons represent forested areas where various resources and processes are at work, including logging (e.g., conventional or reduced impact logging RIL [69]) and mining operations, road construction, and wildlife habitat conservation advocated by NGOs. Adapted with permission from Ref. [24].

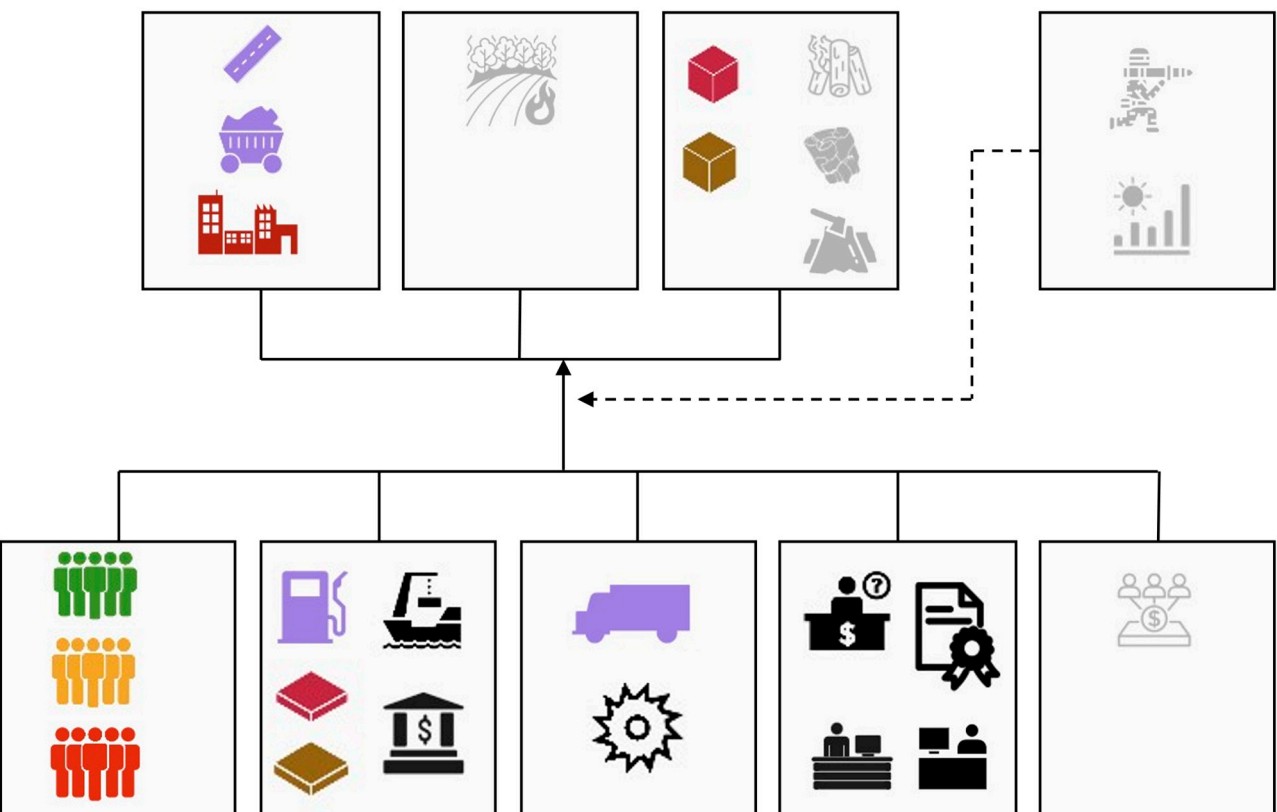

**Figure 2.** Drivers of forest change. The conceptual model used in MineSet for analyzing drivers of forest change is based on Geist and Lambin's [70] framework. This model serves as a foundation for the international forestry course and provides a platform for engaging discussions. The framework consists of proximate causes (top)—infrastructure extension (box 1: street, mining, and settlements), agricultural expansion (box 2: slash and burn cultivation), wood extraction (box 3: red and brown wood logging, illegal logging, firewood, and charcoal), other factors (box 4: war and climate change); and ultimate causes (bottom)—demographic factors (box 1: population growth and migration), economic factors (box 2: gas prices, red and brown wood sawn boards' production, European and Asian markets, and banks), technological factors (box 3: transportation and sawmills), policy and institutional factors (box 4: credit ratings and certificates (e.g., FSC), Ministry of Mining office; Ministry of Forestry office), cultural factors (box 5: public relation and shareholder values). Game tokens are represented in color; roles performed by the facilitating team are in black; game actions or events are in grey. Icons source: Flaticon.com.

Central to the MineSet model is the process of forest growth and the interaction between ecological processes and human activities. Each cell has a value of forest cover (F) ranging from 0 to 10, represented visually with a different color according to three broad landcover types (in the game, dark green, light green, yellow) [24]. In addition, each cell has a Maximum Forest Cover (Fmax) also ranging from 0 to 10. F cannot exceed Fmax. Roads, local populations, and mines reduce Fmax, while logging directly reduces F without affecting Fmax. F will increase by 1 unit every turn, up to Fmax. Plantations and silvicultural practices will double the rate of forest growth. Players embark on a journey of discovery, gradually unravelling and comprehending the intricate mechanisms of the system as they progress through gameplay. The MineSet model incorporates mechanisms that simulate the processes of deforestation, forest degradation, natural growth, and restoration within the game. With these simplified rules, the model effectively reflects these four essential processes [24].

Biodiversity is explicitly taken into account in the model through the existence of endemic species represented by tokens located on specific cells. Each of these biodiversity

tokens can be in three states as per the IUCN Red List terminology: least concerned, threatened, or extinct. Based on the land cover type of the cell the species token is placed on, the token will shift from one state to the other. Transitions are reversible—if concession holders decide to invest in reforestation activities—except for the last one, where an extinct species is permanently lost.

Realism in modeling involves accurately representing causal connections and real-life outcomes [71]. Strategy games like MineSet may not be realistic in terms of depicting real-life outcomes but are allegorical representations [23]. However, if realism is defined as aligning participants' understanding with the game's causal structure, games like MineSet are considered realistic. Validating game design is crucial, ensuring it captures real-world constraints and opportunities. Early player feedback—stakeholders who are engaged in co-development and verification of the game to ensure elements and their interactions are well calibrated—helps ascertain alignment with reality [26]. Failure to achieve validity compromises credibility and may lead to misguided conclusions. Participants should maintain critical distance, recognizing model limitations [23]. Teaching game rules and facilitating understanding allows participants to assess alignment with their understanding. Discrepancies highlight diverse perspectives, emphasizing the need for ongoing discussions and collaboration. MineSet has been field tested, calibrated, verified, and validated in a series of workshops with a range of different stakeholders [23–25].

Our approach involves allowing participants to experience the game firsthand, focusing on their individual experiences. The pivotal aspect of the educational process lies in the subsequent debriefing sessions, where in-depth discussions take place. As stated by [55], "learning begins after the game ends." These debriefing sessions serve as the critical moment for reflection and learning. The debriefings occur after each game round and typically extend over several hours. These debriefings involve group discussions that are facilitated by the teachers. To enhance the discussions, an online whiteboard tool called Mural is employed. It serves as a collaborative platform where participants can share their impressions, experiences, and engage in exercises that connect the gameplay to evidence-based and scientific articles. The use of Mural aids in documenting and visually capturing the key points discussed during the debriefings.

## 4. Results

### 4.1. Applied Strategies for Forest Landscape Management

The drivers of deforestation observed in the MineSet game included unsustainable logging operations and agricultural expansion represented as migrants' and settlers' tokens, proximity to international markets (capital-driven deforestation), and insufficient capital (poverty-driven deforestation). By playing the game, the students can gain a deeper understanding of human behavior and agency as factors that shape the system (Figure 2). During the game, the participating students had the opportunity to directly observe and experience the changes in the forest landscape in Central Africa (Figure 3).

The students who played MineSet made a series of critical decisions that shaped the gamescape (game landscape, Figure 3) and social and economic outcomes. Their strategies varied from intensive logging and infrastructure construction in the 1960s to prioritizing sustainability and managing the environment in the 2010s. Each decade presented unique challenges and opportunities that required different approaches to earn money from logging while protecting the environment and satisfying stakeholders.

In the 1970s, players diversified their approaches to earning money from logging by conducting scientific assessments, inventories, and cooperating with the public and government. They expanded harvest volume and roads, sold wood to the international market and shareholders, and built only one road and bought only one truck to gain access to the European market.

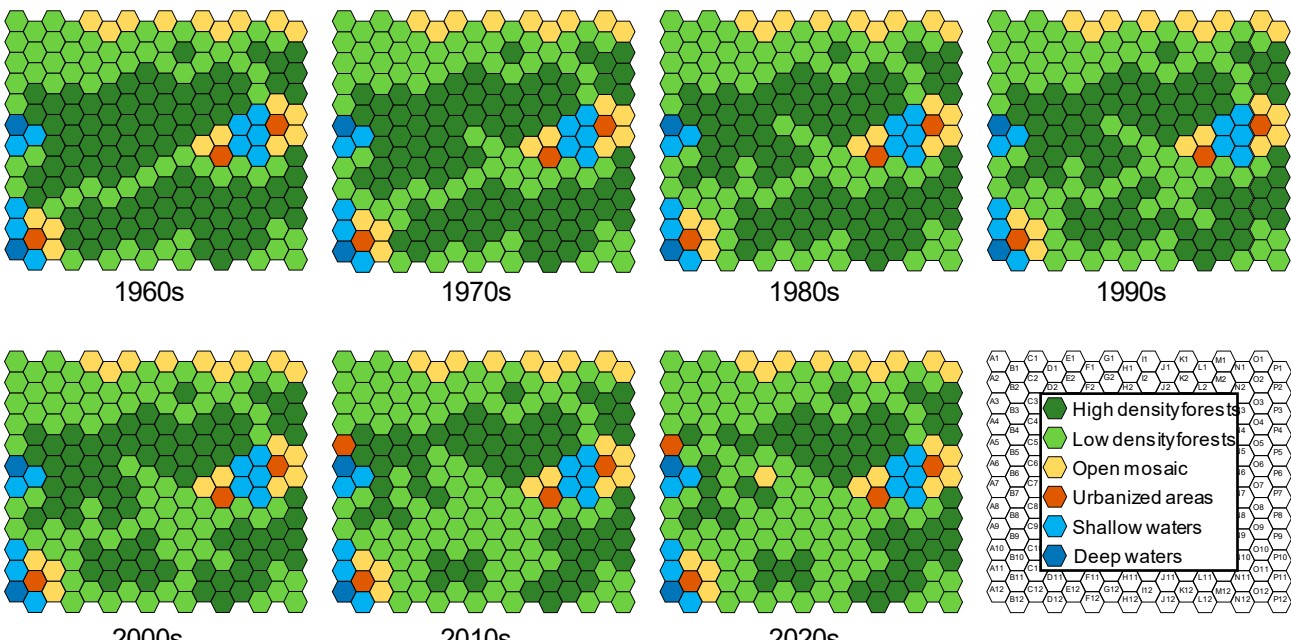

**Figure 3.** Landtype changes from the 1960s to the 2020s of the gamescape. Each game round represents a decade and showcases the cumulative outcome of all landtype-based decisions made by the players during that round. If a concession, composed of seven hexagons (boundaries not shown here), remains unutilized in the game, it is likely to still consist of high-density forests (dark green). However, the presence or absence of roads passing through the area also impacts the actual condition of the forests. Note: roads are not depicted in the figure. However, in the game, roads are represented as tokens. Players strategically place these road tokens (cf. Figure 2) to enhance the transportation of timber or ore to the city and markets, aligning with their individual objectives. The terms like high density forest, low density forest, and open mosaic (land) follow the terminology and definition of the High Carbon Stock Approach [72].

The 1980s saw players move to more profitable markets that required certification such as FSC or PEFC. They intensified wood extraction by building additional roads and buying more trucks, controlling migrant groups, and only used existing infrastructure. Investing in a sawmill to sell sawn wood to Europe was also a common strategy.

In the 1990s, players started processing wood at the sawmill and selling sawn wood to the international market. They pursued certification by presenting a management plan, conservation areas, reduced harvesting (e.g., by use of reduced impact logging), and resolving problems with the local community to obtain an FSC certificate. Sustainable management, biodiversity inventory, and endangered species protection were also emphasized.

The 2000s saw players switch to the domestic market and harvest more red wood—where available—for revenue. They returned to the international market after fulfilling FSC certification guidelines, constructed a sawmill to resolve local community issues, and implemented sustainable harvesting overall. Some players went back to the local market with sawn wood and earned good money. Others switched to alternative markets, invested in biodiversity inventory, and thought about either following the "extract and expand" approach or aiming at certification in the case of growing external public pressure (either shareholders or credit rating system or international press).

In the 2010s, players prioritized the sustainability factor as resources continue to be depleted, protected endangered species, and reduced infrastructure expansion. Some could not make any decisions due to the demand for certification, while others took only what was needed, managed the environment well, and sold wood to the local market.

*4.2. Experiencing the Consequences of the Applied Strategies: Lessons Learned by Students*

The changes directly observed included the loss of tree cover, degradation, and fragmentation, which ultimately led to the replacement of high density forests by low density forests or even fully degraded forests (open mosaic or land) (Figures 3 and 4). Students highlighted that "managing tropical forest landscapes is a highly complex endeavor that entails making challenging decisions. (. . .) I see that stakeholders involved in forest management possess diverse interests and goals. (. . .) which can result in either working together or having conflicts" (student 1).

Changes that players were able to detect indirectly or with a time delay, related to the economy. Logging corporations in the game made a steady profit by selling brown and red wood (Figure 5). However, this profit came with significant environmental consequences, such as a growing list of threatened species (Figure 6), growing migrant population, fluctuating public opinions, and unstable market prices. One student realized that "(. . .) communication with other stakeholders is an important aspect in forest conservation which I failed to do in the MineSet game since I was completely focusing on earning profit" (student 5).

During their decision-making process in the MineSet game, students experienced different drivers of deforestation and forest change. "Unsustainable logging operations, combined with road expansion, liberal granting of concessions, and poor binding forestry regulation, alongside agricultural expansion driven by migrant settlers with underlying foreign conflicts" (student 3); "impact of industrial logging and infrastructure development, such as roads and sawmills, (. . .) improving market access. These approaches were driven by specific demands, particularly economic development" (student 1). Student 6 expressed concerns about uncontrolled logging activities driven by the aim "to grow my company and generate profit. . .The lack of restrictions, controls, and punishments allowed for unchecked actions without certification". These drivers were primarily motivated by specific demands such as economic development, gaining access to the European or Asian market, and growing as a company to earn more money. Some students also experienced uncontrolled logging activities with no restrictions, controls, or punishments, and no certification was in place to regulate these activities at certain times and in some of the concessions.

The MineSet game experience of the students highlights the impact of policies, certification, and public expectations on the decisions made by forest players. Certification schemes emerged in the 1990s and played a crucial role in determining market access and profitability for forest players. Complying with certification requirements was found to be challenging, as some players found it too time-consuming or ineffective. "Complying with certification requirements on my concession proved to be incredibly challenging due to the nearby industrial mining operations. The close proximity of these operations made it impossible for me to safeguard the necessary concessions required for certification" (student 3).

New agencies also affected forest governance and management, but player participation varied. Biodiversity commitments impacted forest conservation and restoration, but some players prioritized economic returns. Some players demonstrated social and environmental responsibility through certification, while others opted for non-certified markets. "I had no money to invest in Certification, so I had to go back to the local market to sell my wood" (student 6).

The absence of strict forest policies allowed some players to prioritize full business efficiency and even consider corrupt practices. However, players faced demands for sustainability efforts and certification, especially on the European market, leading some to seek access to alternative and less regulated or restrictive Asian markets [73]. "Demand for certification on European market pushed me towards Asia as sustainability efforts of other players seemed to be connected with too much effort and no subsidies while transitioning, so enough equipment with (bank) capital was also necessary" (student 3).

The students' experience highlights the complexity of forest management, where players must balance economic, social, and environmental factors while navigating changing policies and market demands.

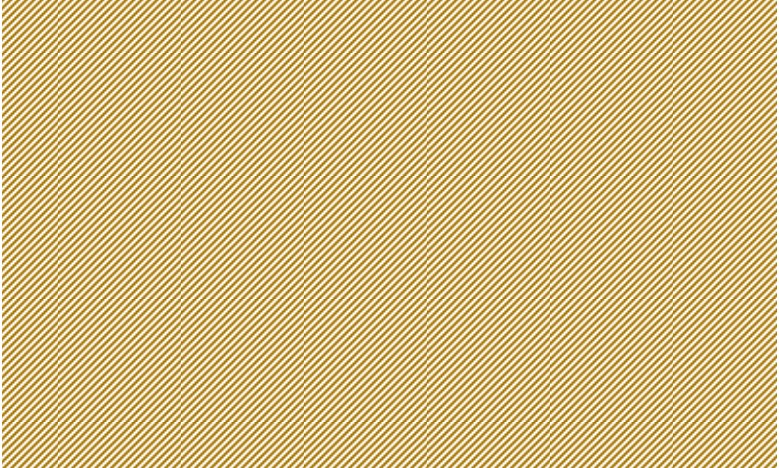

**Figure 4.** Temporal evolution of gamescape landcover types. High density forests (dark green), low density forests (bright green), open mosaic (yellow), and urban settlement (red). Numbers within bars represent the number of units/cells of respective land type.

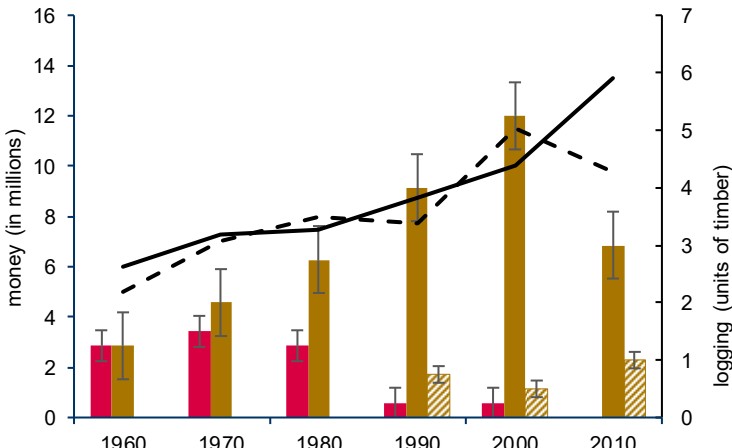

**Figure 5.** Economic dynamics across companies over a 50-year duration. Average costs (dashed line) and earnings (solid line) (left axis); logging activities (right axis); red wood, brown wood, and brown wood planks (hashed bars).

The students emphasized that MineSet, together with the guided support by the lecturers, enable actual change by surpassing solely focused on knowledge acquisition. "(. . .) it was faster to make real learning progress than with conventional lectures" (student 3). In followup discussions with the students, they observed that playing MineSet facilitated them in developing their realistic comprehension of environmental, social, and economic challenges around the topic of deforestation and forest management. It also provided them with a comprehensive understanding of the various factors that shape environmental behavior. "(. . .) it helped me to understand the different perspectives of stakeholders in the game and that will translate in real life situations" (student 1).

The students highlighted that MineSet game simulations cultivate problem-solving abilities that actively encourage sustainable behavior. "(. . .) the game is very illustrative that it allowed me to think and interact to find solutions to address problems" (student 5); "Such games are a great tool to demonstrate the long-term impacts of our decisions in a short time. The game gives space for timely re-evaluation and adjustment of strategies and highlights the complexity of interactions between all stakeholders involved" (student 4).

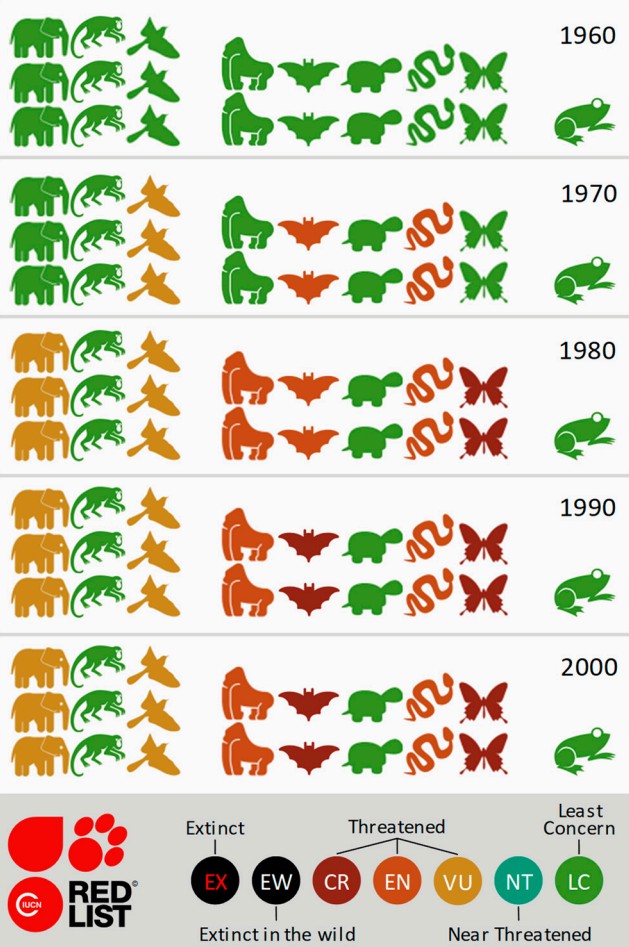

**Figure 6.** Biodiversity. Endemic animal species in the game, their relative richness, and threat status according to the IUCN Red List classification [74].

While the game does not explicitly include tourism, it can be easily adapted to accommodate such activities if players find them valuable or necessary. However, in practice, students made decisions to reduce logging intensity by switching from conventional to reduced impact logging (i.e., taking out only one instead of two brown wood per harvesting activity), or they chose to restrict logging in specific locations to protect biodiversity and maintain forest connectivity (e.g., see the high density forest hexagons in Figure 3).

*4.3. Student Feedback on the Game*

The original MineSet game is a physical haptic tabletop game, which has the advantage of facilitating player-to-player exchanges and emotional engagement. However, for this particular instance of the game, the students played an online version. While the online version allowed for remote play and accessibility, it lacked the tactile and emotional aspects of the physical game. The students offered valuable feedback on how the MineSet game could be improved for educational purposes. To enhance the overall learning experience, they suggested two key changes. Firstly, the students recommended the development of a fully standardized online version of the game, which would be optimal for remote play. While the beta-run of the game on Mural was generally successful, waiting times increased as the game master had to keep track of everything, and instructions for online play took some time for everyone to understand. A standardized online version would also reduce the need for a game master and allow students to proceed through the game at their own pace. Secondly, the students suggested providing an instruction sheet before the start of the game to provide a clear impression of the game mechanics. This would

reduce the time needed for the game master to explain the game, increasing the efficiency of the game, and improving the overall learning experience. The students also highlighted the importance of improving the accessibility and connectivity of the game, which would allow more participants to engage with the game remotely.

These suggestions aim to maximize the educational benefits of the game, facilitating its integration into teachers' curricula and enhancing students' understanding of forest governance and management. Specifically, the game can help students grasp complex dynamics such as deforestation, sustainable forest management, and the intricate interplay between society, the state of nature, and power asymmetries among various actors.

Throughout the game, the students demonstrated a wide variety of strategies. Over the span of 60 years, players had the freedom to make choices that aligned with their priorities. They could prioritize intensive logging to maximize timber utility or consider additional values such as biodiversity and social aspects. The MineSet game provided an open-ended solution space, allowing for a diverse range of strategies to be explored.

During the debriefing and discussion sessions that took place outside of the game-play, we carefully evaluated these strategies through what-if scenario discussions. These discussions took into account the specific context and situation of each player's decisions. The main objective was to gain a deeper understanding of the players' choices, their motivations, and the potential implications of their decisions. By engaging in these discussions, we aimed to enhance the learning experience and facilitate critical thinking among the students.

## 5. Discussion

### 5.1. Transformative Learning

In the past decade, there has been a growing interest among educational researchers in transformative experiences. Kevin Pugh [75] has recommended that science classrooms should be reoriented to incorporate such experiences, while transformative learning theory and social justice education have emphasized the significance of creating "disorienting dilemmas" to transform students' worldviews [76]. In the context of this discussion, during their gameplay of MineSet, the students were presented with a few critical aspects that were surprising to them. Firstly, they were taken aback by how their collective decisions had a negative impact on the forests, which helped them comprehend that managing tropical forest landscapes is a multifaceted task that involves difficult choices. Secondly, the students realized that unexpected events and feedback could significantly influence the system and outcomes. Lastly, they came to understand that stakeholders may have different interests and goals, which can result in either cooperation or conflicts. These surprising elements of the game underscore the value of games like MineSet, which provide a realistic and immersive experience that can prepare students for real-world scenarios in forestry and environmental contexts.

Transformative learning is a process that involves critically evaluating one's existing assumptions and beliefs to bring about a profound shift in worldview [77]. This transformative process is facilitated by being exposed to alternative viewpoints and engaging in conscious reflection. In the context of higher education, transdisciplinary approaches have the potential to foster transformative learning by exposing students to diverse knowledge and perspectives. These approaches emphasize reflexivity, encouraging students to unpack their values, norms, and worldviews as they grapple with sustainability issues. However, there is an ongoing discussion on how to effectively design transdisciplinary learning experiences that optimize transformative learning. Education research recognizes the value of transformative practices in sustainability sciences, but there is a dearth of studies investigating their integration with learning in the field. Participatory pedagogies promoting critical self-reflection are crucial, yet educators face barriers including awareness gaps, resistance, limited funding, inadequate rewards, individualistic research approaches, and bureaucratic hindrances to integration efforts [78–80]. In our observations, we have found

that strategy games have proven to be particularly effective in promoting transformative learning within transdisciplinary contexts.

Optimal learning occurs when individuals actively engage in learning practices that are directly relevant to their specific situations. These practices are primarily influenced by unconscious processes in the brain, by our mental models, shaped by previous learning experiences and motivational states [81]. In the realm of higher education, achieving transformative learning often requires the participation of stakeholders [82,83]. However, incorporating real stakeholders into the classroom is hindered by practical challenges, including scheduling conflicts, time constraints, and other commitments. While virtual platforms provide an alternative, organizing stakeholder engagement remains a complex endeavor. To overcome these obstacles and emulate the inclusion of stakeholders, the integration of role-play games emerges as an elegant solution.

Role-play games like MineSet fulfill the four criteria outlined by Chen and Martin [84] for an effective transformative environmental education approach. Firstly, they surpass mere knowledge acquisition by aiming to facilitate actual change, such as the management of the forest landscape within MineSet. Secondly, role-play simulations enable participants to understand and experience environmental challenges in a real-world context. For example, players' strategies in MineSet manifest as changes in land types, impacting factors like migration, productivity, and biodiversity. Thirdly, these simulations shed light on both internal and external influences on environmental behavior, providing a comprehensive understanding of the factors at play. In MineSet and the classroom, the central framework focused on direct and indirect drivers of forest change by Geist and Lambin [70]. Lastly, role-play simulations foster a problem-solving approach, requiring participants to actively seek solutions. In MineSet, players typically shifted away from unsustainable logging practices, pursued certification, engaged in agreements with conservation NGOs, or initiated community-based projects. By fulfilling these criteria, role-play simulations like MineSet have the potential to promote transformative learning outcomes and encourage sustainable behavior in the context of environmental education.

Through playing as CEOs and interacting with diverse actors and roles in the game, students actively participate in the game and gain first-hand experience of stakeholder perspectives, thereby fostering a deeper understanding of diverse viewpoints. This immersive approach enables them to navigate the complexities of decision-making within forest landscapes. Throughout the gameplay, students interacted with a range of actors, including big logging companies, concession holders, indigenous communities, government officials, and non-governmental organizations, other land users and nature protection groups. These interactions provided opportunities to exchange perspectives on deforestation and gain a comprehensive understanding of the complex dynamics at play. Importantly, the game also facilitated timely reassessment and adjustment of strategies, underscoring the intricate web of interactions among stakeholders involved in specific issues.

To evaluate the effectiveness and scope of substituting actual stakeholder involvement with role-play games, further scientific research is essential. Nonetheless, the incorporation of role-play games in educational settings signifies a significant advancement towards fostering transformative teaching practices. By immersing students in dynamic scenarios and encouraging active engagement, these games promote critical thinking, empathy, and a holistic understanding of real-world challenges [85]. Ultimately, the inclusion of role-play games offers a promising avenue to bridge the gap between theory and practice, equipping students with the skills and perspectives necessary to navigate complex issues and contribute to meaningful change in the world.

Through the game, students were introduced to the drivers of deforestation and gained a new appreciation for the impact that individual decisions can have on the collective outcome. They learned how different actors in the system have different incentives and interests and how these can lead to tradeoffs between environmental, economic, and social objectives. By exploring these drivers and tradeoffs through the game, students gained a deeper understanding of the challenges associated with managing forest landscapes.

Tradeoffs emerged as a critical concept in the MineSet game. Students learned that managing forest landscapes involves difficult choices and that there are no easy solutions. They gained insights into the interplay between environmental, social, and economic objectives and saw how these can sometimes conflict with each other. By experiencing these tradeoffs firsthand, students were better equipped to understand the complexity of forest management and to appreciate the importance of balancing competing interests.

Constructivist teaching is centered on the belief that learners in general actively construct knowledge rather than passively receive it [86]. It promotes critical thinking skills, nurtures self-motivated and independent learners, and recognizes the dynamic interplay between internal factors and external conditions [87]. This approach emphasizes five key characteristics: active, constructive, self-directed, social, and situational learning processes [85]. Knowledge is constructed within various contexts and through social interactions. Learners co-create knowledge in a self-organized manner, benefiting from complex and authentic learning environments that encourage experiential learning, multiple perspectives, social collaboration, and instructional support [88]. Gaming simulation aligns well with constructivist theories, challenging traditional notions of knowledge and supporting diverse perspectives [85]. It embraces the idea that there is no singular best approach to learning, echoing the principles of constructivism and systems thinking, which encourage the exploration of complexity and reject simplistic solutions in understanding human behavior [85].

*5.2. When Teachers Become Facilitators*

The impact of teachers' actions in the classroom on student learning is widely acknowledged. However, there is a lack of data regarding which teaching practices effectively support learning in inclusive classrooms, particularly data derived from direct observations of teachers [89]. When integrating gaming into the classroom, the teacher's role shifts to that of a facilitator or game master, extending beyond traditional curriculum instruction. During gameplay, various emotions may arise, demanding immediate attention [90]. The teacher must possess the facilitation skills to adeptly manage these strong emotions stemming from both successful and frustrating gameplay. Emotions experienced by participants can span from joy and happiness to frustration, sadness, or even anger, influenced by their decisions and outcomes within the game. The teacher or facilitator must assess the situation and create opportunities for meaningful discussions, often accomplished through debriefing sessions that are integral to the learning process [55]. While debriefings usually follow a structured format [91], in the classroom, they can encompass specific topics or themes. After each round of gameplay in MineSet, which spanned several hours, we would conclude for the day and engage in debriefing sessions for several hours. The topics discussed during debriefing were based on the themes experienced and observed during the gameplay. These discussions drew upon in-game conversations, scientific literature, and evidence. While the students participating in our MineSet gameplay remained in their designated roles as logging and mining companies, we also engaged in discussions where we explored hypothetical scenarios and considered different perspectives. These discussions raised the possibility of role swaps and the implications they might have for real stakeholders, including the potential for participants to take on roles that differ or even oppose their usual real-life roles. The topics covered various aspects such as logging techniques (conventional vs. reduced-impact logging), biodiversity (species loss, species richness, endemicity, IUCN Red List), migration (ethnic diversity, livelihood needs, power asymmetries), certification (FSC and PEFC), markets and permits, as well as transparency and accountability. All students actively participated in these discussions, facilitated to ensure equal engagement. To aid the discussions, we utilized Mural for exercises and visual representation of facts and figures.

Typically, a debriefing consists of several steps, inter alia: (1) acknowledging explicit emotions and inquiring about participants' feelings, (2) discussing the events that unfolded during the game, (3) drawing comparisons between the game and real-world situations,

and (4) what could be changed, i.e., the "what if" discussion. To ensure fruitful outcomes from the game and ensuing discussions, skilled facilitation becomes imperative. While the game sessions can be enjoyable, the process of confronting one's cognitive limitations and challenging incorrect assumptions may lead to frustration and discomfort [92]. Effective facilitation thus plays a crucial role in transforming these challenging personal experiences into valuable opportunities for learning and self-reflection [23]. This approach allows students to co-learn or develop soft competencies which have been identified as key for workplace success. The soft skills development has high importance in our class and includes leadership and management, human relations, and communication.

When organizing a game workshop centered around games like MineSet, it is crucial to acknowledge the inherent risk that players may learn strategies that primarily benefit themselves but have negative consequences for the environment or other stakeholders. Garcia et al. [23] highlighted these risks by emphasizing the connection to power dynamics and asymmetries and conflicts. It is crucial to acknowledge that ecosystems, as voiceless stakeholders, can be impacted in this context. The distribution of responsibilities among the participants, designers, and facilitators is uneven. Gaming organizers hold the responsibility of upholding ethical principles, including competence, integrity, responsibility, respect, and dignity, throughout the design and gaming sessions [23]. Consequently, conducting comprehensive discussions about players' decisions and motivations during the game is indispensable. In the classroom, understanding why students made specific choices and why they did not opt for alternatives is vital for analysis. For instance, we observed students who abstained from logging in a round due to concerns about deforestation and their curiosity about the consequences. In contrast, others acknowledged the impacts of their decisions but prioritized the company's financial performance over other factors. Each round in MineSet centered around specific themes like conservation, restoration, and certification. These themes fostered important "what if" discussions during the debriefing phase.

*5.3. Limitations*

In contrast to many other serious games designed for educational purposes, MineSet distinguishes itself by not having pre-defined winning conditions. Instead, the players, who are students in this context, must make their decisions based on a variety of factors. Firstly, they consider the observable changes occurring within the game environment, particularly on the game board. Secondly, they take into account their performance metrics, such as the quantity of wood they have sold in the markets and the financial status of their company at the end of a game round. Lastly, they factor in the actions and decisions made by other players. As a result, the strategies adopted by players can take numerous forms, leading to a broad and open-ended solution space (refs. [93,94] discuss the significance of considering solution space in role-playing games). Furthermore, it is important to note that each individual player determines their own winning conditions in MineSet. This particular game setup poses challenges when it comes to assessing the impact of the game on educational outcomes. Therefore, it is crucial to emphasize that when games like MineSet are employed in a teaching context, they require careful facilitation and extended debriefing periods, which is at least as important as the gaming itself [55]. These debriefing sessions serve the purpose of establishing meaningful connections between the events and experiences that occurred within the game, validated educational material, and published research outcomes. By engaging in these comprehensive debriefing discussions, educators can better assess the educational impact of MineSet, ensuring that it is leveraged to its full potential for effective learning.

One can improve a game in two ways: internal validity and external validity. Internal validity pertains to how the game functions in isolation, without any external information. This is typically tested with individuals who have no prior knowledge of the game. Ideally, improvements focus on enhancing the realism and scope of the game, which relate to external validity. To achieve this, it is crucial to involve individuals who possess knowledge of both the game and the real-world context. While students can be included in this process,

it requires prior study and familiarity with the subject matter. As students are not experts, their primary role is to engage in the game, raise questions, and then conduct further research by consulting the relevant literature and evidence. They subsequently bring their findings back to the class. It is important to note that the primary beneficiary of this approach is the student, as the ultimate goal is to facilitate their learning experience rather than solely focusing on improving the game itself, which aligns with the original purpose of the class teaching.

*5.4. Using Games beyond the Classroom*

The students provided insights on the usefulness of games like MineSet beyond teaching. As future managers and decision makers in the fields of forestry and other environmental contexts, they recognized the importance of such games in preparing them for real-world scenarios. While games like MineSet provide an alternative to lecture-based learning, the students believed that bringing real actors such as politicians, businesspeople, or locals to the table could further support real action and policy. Moreover, through the possible role swaps the game promotes empathy, which could help overcome opposition in politics or culture, leading to more informed decision making [61].

MineSet can be a useful tool for exploring the transitions and future of tropical forest landscapes due to its ability to simulate the direct and indirect drivers of forest change, as well as mimic forest degradation and recovery. What makes the game particularly valuable is that it brings agency to the table [23]. Real people, such as students or anyone who commits their time and energy to attend a gameplay workshop, can enter different roles, such as big corporations with the power to make landscape-level and long-term impacts, governments, the World Bank, communities, NGOs, and more. By allowing players to step into these roles and interact with each other, the game creates a realistic and immersive experience, as in "this feels real" (see [95]), that can help players understand the complex dynamics of tropical forest landscapes [68,96].

Ultimately, the MineSet game can help players develop a better understanding of the challenges and opportunities involved in sustainable forest management. By exploring different scenarios and outcomes, players can gain insights into how different decisions and actions can affect the landscape over time in a safe space [66,97]. This can be particularly useful for policymakers, as they can use the game to test different policy options and evaluate their potential impact before implementing them in real life [23].

Education possesses the potential to effectively engage and raise awareness among diverse individuals, ranging from ordinary citizens and students to educators and high-level decision makers. However, it is crucial to recognize that information alone is insufficient to instigate significant change [98,99]. While reinforced learning entails gradual shifts in behavior or improvements in performance, insight learning manifests as sudden and dramatic behavioral changes [100]. Surprises, arising from disparities between incoming information and our established expectations formed by mental models, prompt specific behavioral responses aimed at revising these mental models to enhance our ability to anticipate future states of the world [101]. As players engage in gameplay, they undergo emotional responses, and the impact of their decisions shapes their values and priorities. It is this transformative power of games that makes them effective tools, such as MineSet, not only within classrooms but also in realms like policy making [23]. While we have provided an example specifically related to forests, it is important to note that the application of such games can be extended to any complex subject within the realms of environmental management and governance. This includes internationally accepted solutions pertaining to various aspects such as the Convention on Biological Diversity (CBD), forest certification, and more. These games serve as valuable tools for exploring and understanding the intricacies of different environmental issues and their associated management and governance challenges.

*5.5. Towards Sustainable Forestry: Critical Thinking and Social Innovations*

The simulation game employed in this study spans a 60-year timeframe, with each round representing a decade, starting from 1960 and extending beyond 2020. The game encompasses a wide range of topics, including sustainability, biodiversity, certification, financial crises, war, pandemics, among others. By immersing participants in gameplay, the game offers an interactive and experiential learning opportunity that allows students and practitioners to explore their own perspectives and analyze the intricate factors influencing international forest governance. Notably, a well-designed and effectively executed game session can have a profound and enduring impact on participants, evoking strong emotional responses [23]. Throughout gameplay, individuals may experience a spectrum of emotions, such as surprise, frustration, triumph, anger, and joy, which contribute to a heightened level of engagement and deeper understanding. Moreover, the topics and discussions that emerge during the game continue to resonate with participants even after the session concludes, thereby enhancing the overall learning experience [55].

Strategy games in forest education provide an innovative tool for fostering sustainability through critical thinking and transdisciplinary understanding. By engaging with real-world scenarios and using creative problem-solving skills, students develop a nuanced understanding of complex sustainability challenges and the interconnections between stakeholders. This approach can promote motivation and collaboration, preparing the next generation of leaders in the sustainability field, yet fostering changes on the ground, for example, social innovations. By promoting collaboration and creativity among students, working together to devise effective solutions for sustainability challenges, students can develop a sense of ownership and agency over these issues [102]. This approach can inspire new ideas and approaches to sustainability, leading to social innovations that benefit communities and the society as a whole. Moreover, by engaging with a range of disciplines and stakeholders, students can develop a more comprehensive understanding of sustainability challenges and the need for transdisciplinary solutions. This approach can help break down silos and promote collaboration across sectors and fields and can be a valuable tool for fostering social innovations [103,104] to address the complex sustainability challenges in forestry.

After playing the MineSet game, students summarized that games like MineSet can be useful for educating and empowering local communities to overcome problems related to sustainable forestry. They also believe that governmental and non-governmental decision makers could benefit from the game's ability to show different perspectives on the same issue. MineSet includes multiple actors and stakeholders, making it suitable for teaching international forestry classes and providing a better understanding of their behavior towards sustainable forestry. The game can prevent conflicts among stakeholders by guiding participants to think and discern without blaming others, thus helping decision makers create informed policies.

Future research endeavors should prioritize the assessment of the impact of games like MineSet on knowledge creation and the enhancement of understanding complex issues, particularly those currently experienced in the context of global deforestation and biodiversity loss. Additionally, it is crucial to explore the integration of other pressing sustainability problems, such as climate change, into these games. However, it is important to note that simply adding a new module to the existing game would not suffice. Instead, the development of an entirely new game would be required. Given that MineSet is a strategy role-playing game based on participatory modeling, addressing a real-world challenge related to climate change and engaging stakeholders would be necessary to co-develop, verify, and validate the game. This comprehensive approach would ensure the game's effectiveness in addressing real-world issues and engaging players in meaningful learning experiences.

## 6. Conclusions

The integration of role-playing and strategy games, such as MineSet, shows promise for enhancing forest education in classrooms. In the context of evolving global conditions and challenges in the forestry sector, innovation in the classroom becomes increasingly important. Forest education plays a crucial role in sustainable forest management and meeting societal demands on forest resources. However, concerns persist regarding the insufficiency and outdated nature of the current forest education. To address these challenges, it is essential to attract talented students to forest programs, provide ongoing learning opportunities, and broaden access to educational materials through online platforms. The MineSet game serves as a valuable tool by offering immersive experiences that provide insights into complex issues like deforestation. By assuming different stakeholder roles and emphasizing tradeoffs in forest management, the game nurtures critical thinking and collaborative problem-solving skills. Additionally, the MineSet game is instrumental in teaching international forestry by illustrating stakeholder behavior and the challenges associated with sustainable forestry. Its play-based design facilitates unbiased thinking and fosters equitable decision making. Incorporating such innovative tools into forest education enhances the training of future forest managers and policymakers, optimizing the contributions of forests to sustainable development. Continued exploration and implementation of these educational games are crucial for overcoming the limitations in forest education, increasing awareness and understanding of forests and related professions. Embracing innovative approaches in the classroom equips students to tackle complex forest issues, paving the way for effective and sustainable forest management practices.

**Author Contributions:** Conceptualization, P.O.W., M.M. and C.A.G.; methodology, C.A.G. and P.O.W.; validation, E.R., L.V.C., R.L., J.R., T.R., M.Z. and O.S. formal analysis, all; data curation, P.O.W., R.L., M.Z., O.S. and C.A.G.; writing—original draft preparation, P.O.W., M.M., E.R., J.R. and T.R.; writing—review and editing, all; visualization, P.O.W., M.Z., O.S. and C.A.G.; supervision, C.A.G. and P.O.W. All authors have read and agreed to the published version of the manuscript.

**Funding:** This research received no external funding.

**Institutional Review Board Statement:** Not applicable.

**Informed Consent Statement:** Informed consent was obtained from all subjects involved in the study.

**Data Availability Statement:** Data sharing is not applicable to this article.

**Conflicts of Interest:** C.A.G. and P.O.W. disclose their shareholding in LEAF Inspiring Change (https://leafic.ch/), a Swiss spin-off of ETH. LEAF provides consultancy services, including the utilization of strategy games, to clients in both the public and private sectors.

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
