# Peer review of "Fostering Innovation, Transition, and the Reconstruction of Forestry: Critical Thinking and Transdisciplinarity in Forest Education with Strategy Games"

_forests, doi:10.3390/f14081646_

Round 1

Reviewer 1 Report

Unfortunately, the article "Fostering Innovation, Transition, and Reconstruction of Forestry: Critical Thinking and Transdisciplinarity in Forest Education with Strategy Games" presented for review does not show signs of a scientific publication. Yes, the authors presented the introduction quite correctly, but there is no research problem, no hypotheses. The study is the observation of how students play, what stages of the strategic game they go through, what problems they have, but nothing comes of it.

Author Response

Reviewer 1.

Unfortunately, the article "Fostering Innovation, Transition, and Reconstruction of Forestry: Critical Thinking and Transdisciplinarity in Forest Education with Strategy Games" presented for review does not show signs of a scientific publication. Yes, the authors presented the introduction quite correctly, but there is no research problem, no hypotheses. The study is the observation of how students play, what stages of the strategic game they go through, what problems they have, but nothing comes of it.

Authors: We understand your concern regarding the absence of traditional scientific elements such as a research problem and hypotheses. It is important to note that this piece deviates from the structure of a classic scientific article, as we have clearly indicated in the Introduction; see lines 93ff. The focus of our work lies in the descriptive nature of the academic contribution, which centers on the observation and analysis of students' gameplay experiences and the juxtaposition of game elements and “personal game experience” with established frameworks such as the one from Geist and Lambin for the drivers of deforestation. While we acknowledge that our approach may differ from conventional scientific methodologies, we believe it provides valuable insights into the practical application of gaming in forest education.

We have attached the revised manuscript in track change version.

Sincerely,

Patrick Waeber
On behalf of the coauthors

Reviewer 2 Report

The article describes a state-of-the-art educational game to support students' comprehensive education and skills in sustainable forest management. Despite the great usefulness and educational potential of the described game, I regret to have to reject the article. Indeed, the mere description of the game and how someone played it is not sufficient for the text to be considered a scientific article. What is missing is a methodology and a framework for determining the effectiveness and value of this experiment from the point of view of the participants as well. Simply citing the literature that games arouse emotions is not sufficient here. What should be investigated are these emotions, feelings, priorities, the change in the level of knowledge and awareness of those students who played the game.

Of the more general issues, I would like to raise two in this introduction. Firstly - I very much miss the clarification in this article (at the beginning), is it about educating future foresters; is it about teaching future foresters how to educate the public (what they can use in this education); or is it about educating the broader public (students from various non-forestry schools) about forestry topics? Without this clarification, the reader may struggle to understand it correctly.

Secondly - Can this game be modified from country to country? Because on different continents and in different countries, there may be different problems and priorities, while educational projects have the best effect when they relate directly to the person involved, as it relates to his or her life and environment.

I provide detailed comments on the text below.

Abstract

There is no information on which age group the game is developed for. There is no information about whether this game has been tested and evaluated by users and whether it has had the effect that was declared by the authors. For me, this is a necessary element to be able to treat this article as a scientific article and not just an advertisement for a product.

Introduction

L40-42 - I think it is worth adding that forest education can also have the effect of increasing public acceptance of forest management activities. This is particularly relevant in more developed societies, where there is particularly high pressure to protect and exclude forests from forest management.

L59-60, L81-82, 90 - unexplained abbreviations are used (immediately after the full name is used, its abbreviation should be given in brackets). The same is true later in the text (e.g. L106, 113) - please correct this everywhere

L68 - "forest education" - whose - future foresters (students) or society in general?

At the end of the Introduction, a paragraph should be added that precisely states the objectives of this article/research, as well as highlighting its importance (the relevance of the research carried out) to the international reader

Materials and Methods

L112-120 - where was this game taken from? Was it developed in some collaboration with people from the region described, or is it purely theoretical (assumptions prepared by authors from outside the region)? How was the balance between development and conservation defined? I mean, how realistic is it, what assumptions have been made about the correct goal/result - these are elements that affect the educational value of the game for university students. This would need to be described in more detail, especially as the whole article is based on this game

L134 - if there are 9 concessions but only 6 students playing, is this not a problem? does this game have to be played by 9 people at once or can the game be played by 1 person? please add this detail

L142 - why and what are the players discussing with each other?

Fig. 1 - not all arrowheads are legible, and this is important for the interpretation of this model. There is an unnecessary enter marker after the word Cells. Caption - L180 - should be 'Cells' not 'Cell' (if it is to be consistent with Fig.1)

L211 - are there any consequences provided for if a concession holder drives a species to extinction? Can these species not be reintroduced?

For example, is there any provision in this game for revenues from tourism and viewing of endangered animals if the concession owner restricts timber harvesting at this location?

What is missing from this chapter is a more detailed description of what the discussions with the students were like, what was the framework adopted by the authors to determine the effectiveness of such an education model - if this is to be a scientific article, such methodological issues must be described. Were the discussions with the students also aimed at improving this game?

Results

L213-222 - you don't leave such 'suspended' texts, it is better to move it to some sub-section.

L213-214 - it should probably be added here that these were the students' choices, as there are more different threats in the game itself

Fig. 3 - is this a picture of one student's game or all six at once? Where is the road mentioned in the earlier description? How does the fact that three people were missing to make all the concessions affect this picture? Are these the areas with the best forests?

L225-227 (and to the end of this subsection) - and could the students have used any other solution in the era than these general assumptions described earlier? e.g. could they have left some of the forest for nature conservation (endangered species) in the 1960s? This should be added in the description of the game in the methodology. If they did not have the opportunity to make other decisions, then the description of the results obtained will be quite similar to the description of the game assumptions and it is difficult to talk about the students' individual strategies here, especially in the early stages of the game when these opportunities and circumstances are more limited. What is missing here is information on what the consequences of using solutions other than those indicated in the era might have been.

L254 - "Some could not make any decisions due to the demand for certification" - is this in line with the game premise from L167-169 (As the game unfolds, players discover the complexity of the system and devise new rules and strategies to balance development and conservation)? Because it doesn't look like a balance. Well, unless these students chose wrongly beforehand and that's why it came out that way - that would require a comment

Subsection 3.1 lacks information from discussions with students - why they made the choices they did and not others etc., how they rate this element of the game (opportunity to build their strategy) etc.

L257 - in what sense of degradation?

L258 - what does 'secondary forest' and 'fully degraded forest' mean in this game? In fig. 2 and 3 there are only two categories of forests with a different name: high density forests and low density forests. These are all different terms, please sort it out and standardise it

L271 – Chinese or Asian?

L283-285 - "The absence of strict forest policies allowed some players to prioritize full business efficiency and even consider corrupt practices." - is this really what the game was supposed to teach?

What were the consequences for the 'businesses' run by the students depending on the strategy adopted? e.g. growth of the business or its bankruptcy (in extreme situations). Such a more direct link between strategy and consequences is missing here

Fig. 6 - broader commentary for it in the text is missing

Discussion

At the outset, I would like to introduce such a doubt - is the presentation of the specifics of forest management from the middle of the last century not "dangerous" from the point of view of teaching students about modern sustainable forest management? I am referring to the aforementioned choice of students who paid attention exclusively to economic issues, including corruption. Was this solution in the game (such a long time perspective including incorrect management) correct? did the discussions with the students ultimately allow to orient their thinking and knowledge correctly? did they suffer any consequences of their incorrect choices in the game?

L302-314 - please put this 'suspended' text in some subsection

L305 - pandemics? there was nothing on this topic before, how was it covered in the game??

L308 - why international? what are the elements of international forest governance here?

L308-314 - it is difficult to confirm this in the case of the game described if the authors did not foresee any system of evaluation of students' knowledge, skills and satisfaction in their work!

L322 - collective? then the students didn't play each individually as a different concession owner? maybe this is about the sum of individual decisions?

L336-338 - well, that's what the educational effect should be (understanding the meaning and complexity of sustainable forestry), hence my doubt about "playing" with the old forest management rules too - what educational effect will this have? lest we conclude that "it used to be simpler, easier, better", so maybe let's do it that way now too

L339-341 - anything more on this? what is the contribution?

L353 - "By assuming different roles" - these roles are not very different, as these are all concession holders. A different role is e.g. a representative of an NGO or a local community

L357-358 - such information should be part of the results

L385-387 - not all (I'm thinking of those who chose to be corrupt)

L401 - citation missing

L403-428, L451-489 - this is theory with no reference to the educational process of these six students, including their direct impressions and evaluation of the game, so should be included in the Introduction or when describing the game in Materials and Methods

L431 - what 'interplayer exchanges'?

L434-447 (and onwards) - such information should be in the results

L450 - but probably also issues of sustainable forest management and its complex relationship with society and the state of nature

L457-459, L464-467 - what role swaps? please compare this with L113-114, 133-134

Conclusions

All these positive comments about the game should be confirmed by the students who took part in these activities, not by the authors of the game.

L520 - forest education of whom? because one can get lost as to whether it is about educating the public or forest managers

Author Response

The article describes a state-of-the-art educational game to support students' comprehensive education and skills in sustainable forest management. Despite the great usefulness and educational potential of the described game, I regret to have to reject the article. Indeed, the mere description of the game and how someone played it is not sufficient for the text to be considered a scientific article. What is missing is a methodology and a framework for determining the effectiveness and value of this experiment from the point of view of the participants as well. Simply citing the literature that games arouse emotions is not sufficient here. What should be investigated are these emotions, feelings, priorities, the change in the level of knowledge and awareness of those students who played the game.

Authors: We appreciate your concern regarding the absence of traditional scientific elements, such as a research problem, hypotheses, assessment grid, and framework in our article. We acknowledge that this piece deviates from the structure typically found in classic scientific articles, as clearly indicated in the Introduction (lines 93ff). Instead, our focus is on providing a descriptive academic contribution that entails presenting and discussing students' gameplay experiences, as well as comparing game elements and 'personal game experience' with established frameworks like Geist and Lambin's model for the drivers of deforestation.

While we understand that our approach may differ from conventional scientific methodologies, we firmly believe that it offers valuable insights into the practical application of gaming in forest education. It is important to note that the game was not originally developed for educational purposes, as clarified in the text. However, we are pioneering its use in a university context and documenting its potential benefits in enhancing learning outcomes.

In the newly added Section 2 (literature review), we provide a clear rationale for why we chose to utilize this game in an educational setting.

Of the more general issues, I would like to raise two in this introduction. Firstly - I very much miss the clarification in this article (at the beginning), is it about educating future foresters; is it about teaching future foresters how to educate the public (what they can use in this education); or is it about educating the broader public (students from various non-forestry schools) about forestry topics? Without this clarification, the reader may struggle to understand it correctly. Secondly - Can this game be modified from country to country? Because on different continents and in different countries, there may be different problems and priorities, while educational projects have the best effect when they relate directly to the person involved, as it relates to his or her life and environment.

Authors: In our abstract, we specifically mention 'graduate students' to highlight our focus on this particular group rather than the general public. Throughout the entire text, we have consistently emphasized our discussion around students. It is important to note that these students represent the potential future foresters and policy-makers, making our research particularly relevant to their educational journey.

Regarding the second point on whether the game can be adapted to other countries and contexts, it is indeed possible. However, it is important to understand the nature of such games. The MineSet game was developed using a participatory modelling approach known as ComMod or companion modelling, with two primary objectives: a) to facilitate collective learning through the co-development, verification, and validation involving relevant stakeholders, and b) to influence policy-making (cf Garcia et al. 2022). Originally developed in the Congobasin (as described in Garcia and Speelman 2017, Dillman et al. 2017), the MineSet game has primarily been used in Central Africa but has also been used in other countries and socio-ecological contexts. The game's design allows players to envision and engage with their own unique world, tailored to their specific context. Stakeholders involved in the development, testing, and validation of the game have found it to be realistic and relevant. The question of whether it would be better to have specific games for specific contexts is a fair and significant one. However, this article's purpose is to clearly outline how we utilize the game with students, enabling them to explore decades of global events. We emphasize the importance of continuous discussions, debriefings, and the incorporation of established frameworks and scientific materials to ensure that students comprehend the significance of their experiences.

I provide detailed comments on the text below.

Abstract

There is no information on which age group the game is developed for. There is no information about whether this game has been tested and evaluated by users and whether it has had the effect that was declared by the authors. For me, this is a necessary element to be able to treat this article as a scientific article and not just an advertisement for a product.

Authors: We have modified abstract, and added a new phrasing, which reads “This paper showcases the transformative use of a strategy game, MineSet, as an innovative teaching method for integrated forest management. The game facilitates deeply engaging experiences that provide unique insights into complex issues like deforestation. By assuming various stakeholder roles, graduate students actively engage with and confront the intricate trade-offs inherent in forest management. This interactive and immersive role-play game not only fosters critical thinking skills but also promotes collaborative problem-solving, making MineSet a highly innovative and attractive tool in forest education. The importance of extended debriefings, facilitation throughout the game, and ongoing discussions should not be underestimated, as they establish meaningful and necessary connections between in-game events, validated educational material, and published research outcomes. Moreover, the game equips students with practical experience and a com-prehensive understanding of landscape approaches, using the Congo Basin as a case study.”

Introduction

L40-42 - I think it is worth adding that forest education can also have the effect of increasing public acceptance of forest management activities. This is particularly relevant in more developed

societies, where there is particularly high pressure to protect and exclude forests from forest

management.

Authors: While we acknowledge the importance of forest education in enhancing public acceptance of forest management activities, backed by extensive scientific literature and several case studies from North America (especially Canada) and Scandinavia, we have made a deliberate decision not to explore this specific aspect in our article. The primary objective and scope of our work lie in equipping future foresters and policy-makers with the necessary knowledge and skills. Nonetheless, we value the reviewer's suggestion and acknowledge the significance of public acceptance, particularly in more developed societies, where there is increasing pressure to balance forest protection with sustainable forest management practices to uphold both the integrity of forests and the reputation of the forestry sector.

L59-60, L81-82, 90 - unexplained abbreviations are used (immediately after the full name is used, its abbreviation should be given in brackets). The same is true later in the text (e.g. L106, 113) -please correct this everywhere

Authors: We have amended as follows: “It was facilitated by the authors CG and PW.” And “Players take on the roles of Chief Executive Officers (CEOs) of logging and mining companies, and they must interact with markets, government, and non-governmental organizations (NGOs) to design strategies and shape the environment, economy, and society with their actions.”

L68 - "forest education" - whose - future foresters (students) or society in general? At the end of the Introduction, a paragraph should be added that precisely states the objectives of this article/research, as well as highlighting its importance (the relevance of the research carried out) to the international reader

Authors: Done. It reads as follows “The objective of this article is to conduct a comprehensive examination of a strategy game that has been developed through participatory stakeholder engagement processes [24,25]. Our focus is on documenting and exploring the game's potential as an education-al tool within the classroom setting. By immersing graduate students in a virtual reality of forest management, the game effectively guides them through evolving challenges, facilitating the development of decision-making skills relevant to real-world forest management scenarios. We provide a detailed account of the game's development, application, practical implementation, and the potential benefits it offers for forest education.

To achieve this objective, our article follows a structured approach. We begin with a brief literature review on the utilization of games in education. Subsequently, we present the game model in the Methodology section, along with its implementation in the classroom. The Results section provides a descriptive account of the gameplay and how the game's outcomes have been utilized for teaching purposes. Finally, we critically discuss the advantages and disadvantages of employing such a game in the classroom, focusing on its capacity to foster transitions, critical thinking, and transdisciplinarity in forestry education.

Materials and Methods

L112-120 - where was this game taken from? Was it developed in some collaboration with people from the region described, or is it purely theoretical (assumptions prepared by authors from outside the region)? How was the balance between development and conservation defined? I mean, how realistic is it, what assumptions have been made about the correct goal/result - these are elements that affect the educational value of the game for university students. This would need to be described in more detail, especially as the whole article is based on this game

Authors: The expanded caption in Figure 1 explains the origin of the game. We made modifications to Figure 1 to enhance the connection between the conceptual model and the final game used in the classroom. Additionally, we added a subsection 3.1 titled “Ensuring the educational value of MineSet for university students”.

L134 - if there are 9 concessions but only 6 students playing, is this not a problem? does this game have to be played by 9 people at once or can the game be played by 1 person? please add this detail

Authors: We state, “Throughout the game, players must compete to acquire logging concessions”, which implies that a logging company can hold more than one concession. To be more explicit, we have modified text, which reads “In total there are 9 concessions, which fall under the jurisdiction of the Ministry of Forestry. The concessions which are located north and south of the already existing road connecting the east and west of the “gamescape” (the game landscape board). Throughout the game, players must compete to acquire logging concessions (during public auctions), construct roads, harvest timber (precious red wood, or common brown wood), and sell their products on the global market.” We have also clarified the number of players: “The game accommodates 5 to 14 players, with players taking on the roles of Chief Executive Officers (CEOs) of logging and mining companies (5 to 7 logging companies in total).” We have further added this sentence: “The game can be played in solitaire, but in that case, the richness of interactions is reduced as all other roles need to be impersonated by the team of teachers.” And we have added this: “These concessions in the game can also be transformed into protected areas by the Ministry of Forestry, introducing concepts such as land sparing, leakage, and strict conservation into the gameplay dynamics.

L142 - why and what are the players discussing with each other?

Authors: We have modified text: “During the initial phase, players have a brief period to strategize and discuss with others—e.g., potential collaboration, infrastructure development, access rights, shared investments, inter alia—before harvesting timber from their concession and sell it to the market.

Fig. 1 - not all arrowheads are legible, and this is important for the interpretation of this model.

There is an unnecessary enter marker after the word Cells. Caption - L180 - should be 'Cells' not 'Cell' (if it is to be consistent with Fig.1

Authors: We have improved Figure 1 and adapted caption text accordingly.

L211 - are there any consequences provided for if a concession holder drives a species to extinction? Can these species not be reintroduced?

For example, is there any provision in this game for revenues from tourism and viewing of

endangered animals if the concession owner restricts timber harvesting at this location? What is missing from this chapter is a more detailed description of what the discussions with the students were like, what was the framework adopted by the authors to determine the effectiveness of such an education model - if this is to be a scientific article, such methodological issues must be described. Were the discussions with the students also aimed at improving this game?

Authors: Biodiversity; we have modified original sentence which reads now “Transitions are reversible—if concession holders decide to invest in reforestation activities—except for the last one, where an extinct species is permanently lost.” We have also added this text to section 4.2.: “While the game does not explicitly include tourism, it can be easily adapted to accommodate such activities if players find them valuable or necessary. However, in practice, students made decisions to reduce logging intensity by switching from conventional to reduced impact logging (i.e., taking out only one instead of two brown wood per harvesting activity), or they chose to restrict logging in specific locations to protect biodiversity and maintain forest connectivity (e.g., see high density forest hexagons in Figure 3).”

Regarding your last question on improving games, we have added the following text to section 5.3.: “One can improve a game in two ways: internal validity and external validity. Internal validity pertains to how the game functions in isolation, without any external information. This is typically tested with individuals who have no prior knowledge of the game. Ideally, improvements focus on enhancing the realism and scope of the game, which relate to external validity. To achieve this, it is crucial to involve individuals who possess knowledge of both the game and the real-world context. While students can be included in this process, it requires prior study and familiarity with the subject matter. As students are not experts, their primary role is to engage in the game, raise questions, and then conduct further research by consulting relevant literature and evidence. They subsequently bring their findings back to the class. It is important to note that the primary beneficiary of this approach is the student, as the ultimate goal is to facilitate their learning experience rather than solely focusing on improving the game itself, which aligns with the original purpose of the class teaching.

In response to your question regarding the framework adopted for assessing the effectiveness of the education model, we did not specifically adhere to a predetermined framework. Our approach involved allowing participants to experience the game firsthand, focusing on their individual experiences. The pivotal aspect of the educational process lies in the subsequent debriefing sessions, where in-depth discussions take place. As stated by Garcia et al. (2016), "learning begins after the game ends." These debriefing sessions serve as the critical moment for reflection and learning. We have added the following text to section 5.3.: “In contrast to many other serious games designed for educational purposes, MineSet distinguishes itself by not having pre-defined winning conditions. Instead, the players, who are students in this context, must make their decisions based on a variety of factors. Firstly, they consider the observable changes occurring within the game environment. Secondly, they take into account their performance metrics, such as the quantity of wood they have sold in the markets and the financial status of their company at the end of a game round. Lastly, they factor in the actions and decisions made by other players. As a result, the strategies adopted by players can take numerous forms, leading to a broad and open-ended solution space ([89,90] discuss the significance of considering solution space in role-playing games). Furthermore, it is important to note that each individual player determines their own winning conditions in MineSet. This particular game setup poses challenges when it comes to assessing the impact of the game on educational outcomes. Therefore, it is crucial to emphasize that when games like MineSet are employed in a teaching context, they require careful facilitation and extended debriefing periods, which is at least as important as the gaming itself [55]. These debriefing sessions serve the purpose of establishing meaningful connections between the events and experiences that occurred within the game, validated educational material, and published research outcomes. By engaging in these comprehensive debriefing discussions, educators can better assess the educational impact of MineSet, ensuring that it is leveraged to its full potential for effective learning.

Results

L213-222 - you don't leave such 'suspended' texts, it is better to move it to some sub-section.

Authors: We have shifted it to 4.1. subsection.

L213-214 - it should probably be added here that these were the students' choices, as there are more different threats in the game itself

Authors: In the game, the students assumed the roles of CEOs and had the authority to make important decisions, including when and where to construct roads and which timber to harvest. However, there was a specific rule that the students were not aware of beforehand: if they opened up forests for logging operations, one of the consequences was the arrival of migrants, represented as tokens in the game. This consequence became evident to the players during gameplay, introducing an element of surprise.

Fig. 3 - is this a picture of one student's game or all six at once? Where is the road mentioned in the earlier description? How does the fact that three people were missing to make all the concessions affect this picture? Are these the areas with the best forests?

Authors: We have modified caption text: “Figure 3. Landtype changes from the 1960s to the 2020s of the gamescape. Each game round rep-resents a decade and showcases the cumulative outcome of all landtype-based decisions made by the players during that round.” A smaller number of players would result in slower change within the game. Concessions fall under the jurisdiction of the Ministry of Forestry. If a concession is not utilized, it is likely to still contain high-density forests, although this depends on whether roads pass through the area or not. Please note that roads are not depicted in Figure 3, but they are represented as tokens in the game. Players strategically place these road tokens to facilitate the transportation of timber or ore to the city and markets based on their own objectives. We have clarified this in caption text of Figure 3.

L225-227 (and to the end of this subsection) - and could the students have used any other solution in the era than these general assumptions described earlier? e.g. could they have left some of the forest for nature conservation (endangered species) in the 1960s? This should be added in the description of the game in the methodology. If they did not have the opportunity to make other decisions, then the description of the results obtained will be quite similar to the description of the game assumptions and it is difficult to talk about the students' individual strategies here, especially in the early stages of the game when these opportunities and circumstances are more limited. What is missing here is information on what the consequences of using solutions other than those indicated in the era might have been.

Authors: In the 1960s, the main objective of the CEOs in the game was to ensure the success of their logging companies by focusing on timber production. The students had the choice to fully embrace the role of a logging company with a primary interest in harvesting timber or to adopt a more cautious approach and wait to see how the situation unfolds. While the majority of students opted for the first strategy, there was one student who decided not to log during the first round. This was discussed during the debriefing sessions. We have added at the end of section 4.2.: “Throughout the game, the students demonstrated a wide variety of strategies. Over the span of 60 years, players had the freedom to make choices that aligned with their priorities. They could prioritize intensive logging to maximize timber utility or consider additional values such as biodiversity and social aspects. The MineSet game provided an open-ended solution space, allowing for a diverse range of strategies to be explored.

During the debriefing and discussion sessions that took place outside of the gameplay, we carefully evaluated these strategies through what-if scenario discussions. These discussions took into account the specific context and situation of each player's decisions. The main objective was to gain a deeper understanding of the players' choices, their motivations, and the potential implications of their decisions. By engaging in these discussions, we aimed to enhance the learning experience and facilitate critical thinking among the students.

L254 - "Some could not make any decisions due to the demand for certification" - is this in line with the game premise from L167-169 (As the game unfolds, players discover the complexity of the system and devise new rules and strategies to balance development and conservation)? Because it doesn't look like a balance. Well, unless these students chose wrongly beforehand and that's why it came out that way - that would require a comment

Authors: Every decision holds significance, including the choice of not taking action. There is rarely a clear right or wrong answer; instead, hindsight reveals more meaningful or less impactful strategies. These discussions took place during the debriefing sessions. While logging and profit were prominent factors in earlier decades of the game (perfectly reflecting the history of global deforestation), the consequences of these decisions emerged in later decades. Additionally, new actors such as NGOs, FSC, and public opinions emerged, which compelled CEOs to re-evaluate their strategies. Section 4.2. shows this, e.g., “The MineSet game experience of the students highlights the impact of policies, certification, and public expectations on the decisions made by forest players. Certification schemes emerged in the 1990s and played a crucial role in determining market access and profitability for forest players. Complying with certification requirements was found to be challenging, as some players found it too time-consuming or ineffective. “Complying with certification requirements on my concession proved to be incredibly challenging due to the nearby industrial mining operations. The close proximity of these operations made it im-possible for me to safeguard the necessary concessions required for certification.” (student 3)”

Subsection 3.1 lacks information from discussions with students - why they made the choices they did and not others etc., how they rate this element of the game (opportunity to build their strategy) etc.

Authors: We have added students experiences and discussions in sections 4.2. and 4.3.

L257 - in what sense of degradation?

Authors: We have clarified this, which reads now “Central to the MineSet model is the process of forest growth and the interaction be-tween ecological processes and human activities. Each cell has a value of Forest cover (F) ranging from 0 to 10, represented visually with a different color according to three broad land cover types. In addition, each cell has a Maximum Forest Cover (Fmax) also ranging from 0 to 10. F cannot exceed Fmax. Roads, local populations, and mines reduce Fmax, while logging directly reduces F without affecting Fmax. F will increase by 1 unit every turn, up to Fmax. Plantations and silvicultural practices will double the rate of forest growth. Players embark on a journey of discovery, gradually unravelling and comprehending the intricate mechanisms of the system as they progress through gameplay. These mechanisms capture the processes of deforestation, forest degradation, natural growth, and restoration in the game. With these simple rules, the MineSet model reflects the four processes of deforestation, forest degradation, forest natural growth, and restoration [24].

L258 - what does 'secondary forest' and 'fully degraded forest' mean in this game? In fig. 2 and 3

there are only two categories of forests with a different name: high density forests and low density forests. These are all different terms, please sort it out and standardise it

Authors: Thanks for flagging this. We have clarified the terms in caption text of Figure 3. “The terms like High density forest, Low density forest and Open mosaic or land follow the terminology and definition of the High Carbon Stock Approach (https://highcarbonstock.org/).” We have also replaced the terms in original L258 accordingly (section 4.2., paragraph 1).

L271 – Chinese or Asian?

Authors: We have changed it to Asian.

L283-285 - "The absence of strict forest policies allowed some players to prioritize full business efficiency and even consider corrupt practices." - is this really what the game was supposed to teach? What were the consequences for the 'businesses' run by the students depending on the strategy adopted? e.g. growth of the business or its bankruptcy (in extreme situations). Such a more direct link between strategy and consequences is missing here

Authors: The game takes players on a 50 to 60-year journey through international forest governance and management, set in a fictional central African country. The students assume the roles of CEOs of international corporations that operate in national and international markets. Their decisions are influenced by public opinion (shareholders, media) and relationships with international donors (banks). The game reflects major global events spanning six decades, starting with a focus on business and timber as core utilities and later incorporating emerging conservation issues, concepts like biodiversity and sustainability (e.g., Rio), and alternatives to conventional management such as certification. The game allows for a wide range of decision-making and exploration of extreme scenarios, such as logging everything versus maintaining intact forests and connecting them across the landscape. These scenarios may have implications for economic sustainability and could potentially lead to bankruptcy. The purpose of the game is to provide opportunities for diverse events and scenarios to unfold, giving players a voice in shaping outcomes. Collective decisions have a scaled impact, while individual decisions may have limited significance in the long run. However, it is crucial for teachers to provide proper facilitation during gameplay, structured debriefings, and in-depth discussions. These elements create an environment where students can ask questions, engage in exercises, and thoroughly explore the various aspects of the game and its consequences. We have ensured that these aspects are in the updated manuscript.

Fig. 6 - broader commentary for it in the text is missing

Authors: Figure 6 serves as an introduction and visual representation of significant topics such as conservation (management approach), biodiversity (concept), the IUCN Red List (tool), and the involvement of the global player, IUCN. While logging and forests are central themes throughout the manuscript, biodiversity is also referenced albeit not as prominently. The purpose of Figure 6 is to provide a broader context and highlight the relevance of biodiversity within the game and its connections to real-world issues.

Discussion

At the outset, I would like to introduce such a doubt - is the presentation of the specifics of forest management from the middle of the last century not "dangerous" from the point of view of teaching students about modern sustainable forest management? I am referring to the aforementioned choice of students who paid attention exclusively to economic issues, including corruption. Was this solution in the game (such a long time perspective including incorrect management) correct? did the discussions with the students ultimately allow to orient their thinking and knowledge correctly? did they suffer any consequences of their incorrect choices in the game?

Authors: Thank you for bringing this up. We have added following text to Section 5.1.: “In our previous work [23], we examined the phenomenon of players learning some-thing that benefits themselves but may have detrimental effects on the environment or other aspects within stakeholder gaming workshops. Garcia et al. [23] emphasized the associated risks by drawing connections to the earlier discussion on power and conflict. It is crucial to acknowledge that ecosystems, as voiceless stakeholders, can be impacted in this context. The distribution of responsibilities among the participants, designers, and facilitators is uneven. Gaming organizers hold the responsibility of upholding ethical principles, including competence, integrity, responsibility, respect, and dignity, throughout the design and gaming sessions [23]. Consequently, conducting comprehensive discussions about players' decisions and motivations during the game is indispensable. Understanding why students made specific choices and why they did not opt for alternatives is vital for analysis. For instance, we observed students who abstained from logging in a round due to concerns about deforestation and their curiosity about the consequences. In contrast, others acknowledged the impacts of their decisions but prioritized the company's financial performance over other factors. Each round in MineSet centered around specific themes like conservation, restoration, and certification. These themes fostered important "what if" discussions during the debriefing phase.

L302-314 - please put this 'suspended' text in some subsection

Authors: shifted to section 5.5.

L305 - pandemics? there was nothing on this topic before, how was it covered in the game??

Authors: Although it was not explicitly covered during the game play, the topic of the pandemic and its impact on global trade, including the tropical timber trade, was discussed. If the game were to continue beyond the 2020s, introducing the concept of a pandemic would be relevant to reflect its influence on the dynamics of global trade.

L308 - why international? what are the elements of international forest governance here?

Authors: The MineSet game, based on the conceptual model shown in Figure 1, incorporates elements of international forest governance, including: International agreements and frameworks: The game addresses forest-related issues, such as deforestation, sustainable management, and biodiversity conservation, aligning with international agreements and frameworks. For example, the Convention on Biological Diversity (CBD) is relevant to these topics, although it may not have been explicitly mentioned in the manuscript. Forest certification and standards: The game recognizes the importance of forest certification and standards, with specific reference to the Forest Stewardship Council (FSC). This highlights the promotion of responsible forestry practices and the assurance of traceability and legality in forest products. Financial mechanisms: The game acknowledges the role of financial mechanisms in supporting conservation and sustainable forest management. This includes activities such as forest restoration and other climate-related initiatives.

L308-314 - it is difficult to confirm this in the case of the game described if the authors did not foresee any system of evaluation of students' knowledge, skills and satisfaction in their work!

Authors: This game has been played extensively beyond the classroom, providing substantial evidence to support this claim. We have added a reference to underpin this statement: #[23]

L322 - collective? then the students didn't play each individually as a different concession owner? maybe this is about the sum of individual decisions?

Authors: MineSet is played by individual CEOs who have the choice to act independently or collaborate with other companies for various reasons, such as outcompeting larger companies in the game. The decisions made by individuals and collaborations during each round have direct consequences on the state of the forests, as depicted in Figure 3.

L336-338 - well, that's what the educational effect should be (understanding the meaning and complexity of sustainable forestry), hence my doubt about "playing" with the old forest management rules too - what educational effect will this have? lest we conclude that "it used to be simpler, easier, better", so maybe let's do it that way now too

Authors: Forest management is indeed a dynamic process that necessitates adaptation to changing conditions. In our international forestry class, whether we use games or traditional teaching methods, we always emphasize the importance of looking into the past to understand the causes, effects, and feedback loops of previous actions. This understanding is crucial in comprehending the current state of forests, including the loss of high-density forests, species decline, increased climate change impacts, and the transgression of planetary and social boundaries. The game serves as educational tool by allowing students to step into the roles of CEOs and experience forest management challenges firsthand. However, it is important to note that game play is just one part of the educational process. Debriefings, facilitated discussions, and the integration of scientific evidence and literature are essential components to ensure a comprehensive understanding of forest change, forest management, and forest governance.

L339-341 - anything more on this? what is the contribution?

Authors: We have rephrased the sentence which reads now “Education research recognizes the value of transformative practices in sustainability sci-ences, but there is a dearth of studies investigating their integration with learning in the field. Participatory pedagogies promoting critical self-reflection are crucial, yet educators face barriers including awareness gaps, resistance, limited funding, inadequate rewards, individualistic research approaches, and bureaucratic hindrances to integration efforts [74–76].

L353 - "By assuming different roles" - these roles are not very different, as these are all concession holders. A different role is e.g. a representative of an NGO or a local community

Authors: We agree. We have changed sentence to “Through playing as CEOs and interacting with diverse actors and roles in the game, students actively participate in the game and gain first-hand experience of stakeholder perspectives, thereby fostering a deeper understanding of diverse viewpoints.

L357-358 - such information should be part of the results

Authors: shifted to Results (section 3.2.)

L385-387 - not all (I'm thinking of those who chose to be corrupt)

Authors: Not sure we fully understand what you mean. Corruption can emerge during gameplay, mirroring its presence in real life. While encountering corruption may not be pleasant, or even worse, it can contribute to players' personal experiences and can serve as a valuable lesson. Again, carefully led discussions during debriefing sessions is key to put also corruption into context.

L401 - citation missing

Authors: Reference added (#82)

L403-428, L451-489 - this is theory with no reference to the educational process of these six students, including their direct impressions and evaluation of the game, so should be included in the Introduction or when describing the game in Materials and Methods

Authors: We kept it in Discussion, for the time being. We added some specifics about our debriefing sessions into that section to make it not only theory-based.

L431 - what 'interplayer exchanges'?

Authors: Sentence changed, which reads now “The original MineSet game is a physical, haptic tabletop game, which has the advantage of facilitating player-to-player exchanges and emotional engagement.

L434-447 (and onwards) - such information should be in the results

Authors: shifted to Results (new section 4.3.)

L450 - but probably also issues of sustainable forest management and its complex relationship with society and the state of nature

Authors: Absolutely. We have changed text to “These suggestions aim to maximize the educational benefits of the game, facilitating its integration into teachers' curricula and enhancing students' understanding of forest governance and management. Specifically, the game can help students grasp complex dynamics such as deforestation, sustainable forest management, and the intricate interplay between society, the state of nature, and power asymmetries among various actors.

L457-459, L464-467 - what role swaps? please compare this with L113-114, 133-134

Authors: While the students in our presented gameplay of MineSet remained in their roles as logging and mining companies, we did engage in discussions where we explored hypothetical scenarios and different perspectives. These discussions raised the possibility of role swaps and the implications they might have for real stakeholders, including the potential for participants to take on roles that differ or even oppose their usual real-life roles. We have addressed this possibility in the Discussion section of the paper.

Conclusions

All these positive comments about the game should be confirmed by the students who took part in these activities, not by the authors of the game.

Authors: We have provided details and quotes of the discussions with students in the Results (section 4.2).

L520 - forest education of whom? because one can get lost as to whether it is about educating the

public or forest manager

Authors: The article specifically addresses forest graduates (students) rather than the general public. This is stated throughout the manuscript.

Reviewer 3 Report

The article has a very large practical dimension. There is a need for more work of this kind, showing the great importance of forest education in shaping appropriate pro-ecological attitudes and promoting the idea of sustainable forest management. Learning through fun, games is a very good method of education regardless of the age of the audience. 

Author Response

Reviewer 2.

The article has a very large practical dimension. There is a need for more work of this kind, showing the great importance of forest education in shaping appropriate pro-ecological attitudes and promoting the idea of sustainable forest management. Learning through fun, games is a very good method of education regardless of the age of the audience.

I recommend publication of the article.

Authors: Thank you for your encouragement. We acknowledge the significant practical dimension of our article. We agree that there is a pressing need for more research in this area, as it demonstrates the vital role of forest education in shaping pro-ecological attitudes and promoting the concept of sustainable forest management. Our study focuses on the utilization of an engaging and educational gaming approach, exemplified by the development of the game MineSet. By presenting MineSet as a tool for fostering critical thinking within the field of forestry, we aim to emphasize the quality and innovative aspects of learning through immersive experience, regardless of the age of the audience. We do so by describing the process.

We have attached a revised version of the manuscript in track change.

Sincerely,

Patrick Waeber,

on behlaf of the authors

Reviewer 4 Report

Article refers to "Fostering Innovation, Transition, and Reconstruction of Forestry: Critical Thinking and Transdisciplinarity in Forest Education with Strategy Games". The article is interesting,  but it is necessary to include these suggestions, i.e.:

1. Please, add in Abstract short sentence about what is oryginality in this study.

2. Introduction is too general and not included enough literature review according to rule "what whas done by other authors about this issue". Please add section "2. Literature review". 

3. What is the main purpose of the study? Please add at the end of Introduction.

4. What is novetly of this approach? Please add at the end of Introduction.

5. What is the research gap? Please add at the end of Introduction.

6. If is possible to create some hipothesis or thesis, please add.

7. Figure 1 is very good and interesting. Good job. I see some highlighting words such as: Hanoverdam/Chindia. Also italic mouse colse to 'Cells". It is possible to remove this highlighting and italic? 

8. Disscussion and Conclusion is very well. 

9. Are some disadvantages of this game/approach? Please add.

10. In Conclusion please add proposition of future research.

Good job! Congratulation.

Author Response

Reviewer 3.

Article refers to "Fostering Innovation, Transition, and Reconstruction of Forestry: Critical Thinking and Transdisciplinarity in Forest Education with Strategy Games". The article is interesting,  but it is necessary to include these suggestions, i.e.:

  1. Please, add in Abstract short sentence about what is oryginality in this study.

Authors: We have modified abstract and added following text “The game facilitates deeply engaging experiences that provide unique insights into complex issues like deforestation. By assuming various stakeholder roles, students actively engage with and confront the intricate trade-offs inherent in forest management. This interactive and immersive role-play game not only fosters critical thinking skills but also promotes collaborative problem-solving, making MineSet a highly innovative and attractive tool in forest education. The importance of extended debriefings, facilitation throughout the game, and ongoing discussions should not be underestimated, as they establish meaningful and necessary connections between in-game events, validated educational material, and published research outcomes.

  1. Introduction is too general and not included enough literature review according to rule "what whas done by other authors about this issue". Please add section "2. Literature review".

Authors: We have introduced a new section, Section 2, which provides a concise literature review. In Section 1, the Introduction, we present the background and global context that necessitate transformative and new approaches in forestry education. In brief, these approaches are necessary to address the concurrent crises and rapidly evolving demands in the field.

  1. What is the main purpose of the study? Please add at the end of Introduction.

Authors: We have added following text at the end of Introduction: “The objective of this article is to conduct a comprehensive examination of a strategy game that has been developed through participatory stakeholder engagement processes (Garcia and Speelman, 2017). Our focus is on documenting and exploring the game's potential as an educational tool within the classroom setting. By immersing students in a virtual reality of forest management, the game effectively guides them through evolving challenges, facilitating the development of decision-making skills relevant to real-world forest management scenarios. We provide a detailed account of the game's development, application, practical implementation, and the potential benefits it offers for forest education.”

  1. What is novetly of this approach? Please add at the end of Introduction.

Authors: To the best of our knowledge, the role-play game we employed, specifically MineSet, which involves a participatory process leading to a tabletop game, has not been utilized in a forestry teaching context, and possibly not in any other teaching context (although this cannot be stated with certainty). It is crucial to emphasize that strategy games effectively portray validated realities, creating a sense of realism that resonates with both players and beta-testers. When immersed in the game's reality, individuals assume specific roles and perceive their experiences as authentic, as demonstrated by Garcia et al. (2022). Therefore, the novelty of this paper lies in its depiction of the utilization of a strategy game within the domain of forestry teaching, highlighting its distinctive contribution. In Section 2, we provide a review of games for education. In the final two paragraphs, we clearly illustrate how MineSet, as a role-play game that was co-developed, verified, and validated in the field with real stakeholders from Central Africa, stands apart from the majority of games used for educational purposes.

  1. What is the research gap? Please add at the end of Introduction.

Authors: The new Section 2 (Literature review) provides an overview of the current state of research on games, specifically serious games, with a particular focus on role-play games in the educational context. In response to comment #4, our article addresses a research gap by utilizing a strategy game that has undergone development, testing, and validation in the field with independent stakeholders. Moreover, MineSet stands out from many other educational role-playing games by offering a broad solution space, where the winning conditions are not pre-defined. This unique aspect enables students to navigate uncertainty, make decisions, observe the consequences of those decisions, experience feedback loops, and adapt their strategies accordingly.

  1. If is possible to create some hipothesis or thesis, please add.

Authors: This article does not test any research hypothesis (see also answer to Reviewer 1). We have added a paragraph that clarifies how this article is structured. “To achieve this objective, our article follows a structured approach. We begin with a brief literature review on the utilization of strategy games in education. Subsequently, we present the game model in the Methodology section, along with its implementation in the classroom. The Results section provides a descriptive account of the gameplay and how the game's outcomes have been utilized for teaching purposes. Finally, we critically discuss the advantages and disadvantages of employing such a game in the classroom, focusing on its capacity to foster transitions, critical thinking, and transdisciplinarity in forestry education.

  1. Figure 1 is very good and interesting. Good job. I see some highlighting words such as: Hanoverdam/Chindia. Also italic mouse colse to 'Cells". It is possible to remove this highlighting and italic?

Authors: Thank you. We have revised the figure to enhance clarity by clearly indicating the representation of different elements from the conceptual model in the actual game. The modifications made in the figure and its caption aim to provide a more straightforward understanding of the relationships between the conceptual model and the corresponding items in the game.

  1. Disscussion and Conclusion is very well.

Authors: Thank you. We have made few modifications in the Discussion section to address comments ##9 and 10.

  1. Are some disadvantages of this game/approach? Please add.

Authors: Section 5.3. Limitations specifically deals with these aspects. We have added the following text “In contrast to many other serious games designed for educational purposes, MineSet distinguishes itself by not having pre-defined winning conditions. Instead, the players, who are students in this context, must make their decisions based on a variety of factors. Firstly, they consider the observable changes occurring within the game environment. Secondly, they take into account their performance metrics, such as the quantity of wood they have sold in the markets and the financial status of their company at the end of a game round. Lastly, they factor in the actions and decisions made by other players. As a result, the strategies adopted by players can take numerous forms, leading to a broad solution space ([89,90] discuss the significance of considering solution space in role-playing games). Furthermore, it is important to note that each individual player determines their own winning conditions in MineSet. This particular game setup poses challenges when it comes to assessing the impact of the game on educational outcomes. Therefore, it is crucial to emphasize that when games like MineSet are employed in a teaching context, they require careful facilitation and extended debriefing periods, which is at least as important as the gaming itself [55]. These debriefing sessions serve the purpose of establishing meaningful connections between the events and experiences that occurred within the game, validated educational material, and published research outcomes. By engaging in these comprehensive debriefing discussions, educators can better assess the educational impact of MineSet, ensuring that it is leveraged to its full potential for effective learning.

  1. In Conclusion please add proposition of future research.

Authors: We have added following text into subsection 5.5 Towards sustainable forestry. It reads “Future research endeavors should prioritize the assessment of the impact of games like MineSet on knowledge creation and the enhancement of understanding complex issues, particularly those currently experienced in the context of global deforestation and biodiversity loss. Additionally, it is crucial to explore the integration of other pressing sustain-ability problems, such as climate change, into these games. However, it is important to note that simply adding a new module to the existing game would not suffice. Instead, the development of an entirely new game would be required. Given that MineSet is a strategy role-playing game based on participatory modeling, addressing a real-world challenge related to climate change and engaging stakeholders would be necessary to co-develop, verify, and validate the game. This comprehensive approach would ensure the game's effectiveness in addressing real-world issues and engaging players in meaningful learning experiences.”

Good job! Congratulation.

Authors: Thank you for your valuable comments. We have taken them into careful consideration and made the necessary revisions accordingly. We hopa that our updated version adequately addresses your concerns.

New literature added:

  • Dillman, C.; Cornioley, T.; Chamagne, J.; Garcia, C. Defining indicators for Intact Forest Landscapes in the Congo Basin using a role-playing game. A joint report ForDev/FSC Congo Basin, Zürich, Switzerland. http://dx.doi.org/10.13140/RG.2.2.29544.16645
  • Étienne, M. (ed.) Companion modelling: a participatory approach to support sustainable development. Springer Science & Business Media.
  • Barreteau, O.; Bousquet, F.; Étienne, M.; Souchère, V.; d’Aquino, P. Companion Modelling: A Method of Adaptive and Par-ticipatory Research. In: Étienne, M. (eds) Companion Modelling. Springer, Dordrecht. https://doi.org/10.1007/978-94-017-8557-0_2
  • Abt, C. Serious Games, New York, Viking Press, 1970.
  • Nazry, N.N.M.; Romano, D.M. Mood and learning in navigation-based serious games. Comput. Hum. Behav. 2017, 73, 596–604. https://doi.org/10.1016/j.chb.2017.03.040
  • Bellotti, F.; Kapralos, B.; Lee, K.; Moreno-Ger, P.; Berta, R. Assessment in and of serious games: an overview. Advances In Human-Computer Interaction 2013, 2013, 1–1. https://doi.org/10.1155/2013/136864
  • Juan, A.A.; Loch, B.; Daradoumis, T.; Ventura, S. Games and simulation in higher education. Int. J. Educ. Technol. High. Educ. 2017, 14, 1–3.
  • Zhonggen, Y. A meta-analysis of use of serious games in education over a decade. Int. J. Comput. Games Technol. 2019, 2019, 4797032. https://doi.org/10.1155/2019/4797032
  • Ullah, M.; Amin, S.U.; Munsif, M.; Safaev, U.; Khan, H.; Khan, S.; Ullah, H. Serious games in science education. A systematic literature review. Virtual Real. Intell. Hardw. 2022, 4(3), 189–209. https://doi.org/10.1016/j.vrih.2022.02.001
  • Greitzer, F.L.; Kuchar, O.A.; Huston, K. Cognitive science implications for enhancing training effectiveness in a serious gaming context. Educ. Resour. Comput. (JERIC) 2007, 7(3), 2-es. https://doi.org/10.1145/1281320.1281322
  • Tobias, S.E.; Fletcher, J.D. Computer games and instruction. IAP Information Age Publishing, 2011.
  • Connolly, T.M.; Boyle, E.A.; MacArthur, E.; Hainey, T.; Boyle, J.M. A systematic literature review of empirical evidence on computer games and serious games. Comput. Educ. 2012, 59(2), 661–686. https://doi.org/10.1016/j.compedu.2012.03.004
  • Wouters, P.; van Nimwegen, C.; van Oostendorp, H.; van der Spek, E.D. A meta-analysis of the cognitive and motivational effects of serious games. J. Educ. Psychol. 2013, 105(2), 249–265. https://doi.org/10.1037/a0031311
  • Arnab, S.; Lim, T.; Carvalho, M.B.; Bellotti, F.; De Freitas, S.; Louchart, S.; Suttie, N.; Berta, R.; De Gloria, A. Mapping learning and game mechanics for serious games analysis. Br. J. Educ. Technol. 2015, 46(2), 391–411. https://doi.org/10.1111/bjet.12113
  • Sabirli, Z.E.; Coklar, A.N. The effect of educational digital games on education, motivation and attitudes of elementary school students against course access. World J. Educ. Technol. 2020, 12(3), 165–178. https://doi.org/10.18844/wjet.v12i3.4993
  • Iten, N.; Petko, D. Learning with serious games: Is fun playing the game a predictor of learning success? Br. J. Educ. Technol. 2016, 47(1), 151–163. https://doi.org/10.1111/bjet.12226
  • Jumaat, N.F.; Tasir, Z. Instructional scaffolding in online learning environment: A meta-analysis. In 2014 International Conference on Teaching and Learning in Computing and Engineering 2014, 74–77. IEEE. https://doi.org/10.1109/LaTiCE.2014.22
  • Zagal, J.P.; Deterding, S. Definitions of "role-playing games". In Role-Playing Game Studies 2018, 19–51. Routledge.
  • Daniau, S. The transformative potential of role-playing games: From play skills to human skills. Simulation & Gaming 2016, 47(4), 423–444. https://doi.org/10.1177/1046878116650765
  • Winardy, G.C.B.; Septiana, E. Role, play, and games: Comparison between role-playing games and role-play in education. Social Sciences & Humanities Open 2023, 8(1), 100527. https://doi.org/10.1016/j.ssaho.2023.100527
  • DeNeve, K.M.; Heppner, M.J. Role play simulations: The assessment of an active learning technique and comparisons with traditional lectures. Innov. High. Educ. 1997, 21, 231–246.
  • Cox, J.M. Role-playing games in arts, research, and education. Int. J. Educ. through Art. 2014, 10(3), 381–395. https://doi.org/10.1386/eta.10.3.381_1
  • Bytheway, J. A taxonomy of vocabulary learning strategies used in massively multiplayer online role-playing games. Calico J. 2015, 32(3), 508–527.
  • Rahman, A.A. and Angraeni, A., 2020. Empowering learners with role-playing game for vocabulary mastery. International JLTER, 19(1), pp.60–73. https://doi.org/10.26803/ijlter.19.1.4
  • Kaylor, S.L. Dungeons and Dragons and literacy: The role tabletop role-playing games can play in developing teenagers' literacy skills and reading interests. UNI Scholar Works. https://scholarworks.uni.edu/grp/215/
  • Reibelt, L.M.; Stoudmann, N.; Waeber, P.O. A role-playing game to learn and exchange about real-life issues. Learn2Change—Transforming the World through Education. R. Hembrom, T. Holthoff, G. Janecki, S. Laustroer, M. Rolle, and L. Zulu (eds.), pp. 163–171.
  • Jang, Y.; Ryu, S. Exploring game experiences and game leadership in massively multiplayer online role-playing games. Br. J. Educ. Technol. 2011, 42(4), 616–623. https://doi.org/10.1111/j.1467-8535.2010.01064.x
  • Watcharasukarn, M.; Krumdieck, S.; Green, R.; Dantas, A. Researching travel behavior and adaptability: using a virtual reality role-playing game. Simulation & Gaming 2011, 42(1), 100–117.
  • Edwards, P.; Sharma-Wallace, L.; Wreford, A.; Holt, L.; Cradock-Henry, N.A.; Flood, S.; Velarde, S.J. Tools for adaptive gov-ernance for complex social-ecological systems: a review of role-playing games as serious games at the community-policy interface. Environ. Res. Lett. 2019, 14(11), 113002. https://doi.org/10.1088/1748-9326/ab4036
  • Lasley, J. Role-playing games in leadership learning. New Directions for Student Leadership 2022, 174, 73–87. https://doi.org/10.1002/yd.20501
  • Kongmee, I.; Strachan, R.; Montgomery, C.; Pickard, A. Using massively multiplayer online role-playing games (MMORPGs) to support second language learning: Action research in the real and virtual world. In: 2nd Annual IVERG Conference: Im-mersive technologies for Learning: virtual implementation, real outcomes, 27–28 June 2011, Middlesborough, UK.
  • Xanthopoulou, D.; Papagiannidis, S. Technol. Forecast. Soc. Change 2012, 79(7), 1328–1339. https://doi.org/10.1016/j.techfore.2012.03.006
  • Whyte, E.M.; Smyth, J.M.; Scherf, K.S. Designing Serious Game Interventions for Individuals with Autism. J. Autism. Dev. Disord. 2015, 45, 3820–3831. https://doi.org/10.1007/s10803-014-2333-1
  • Othlinghaus-Wulhorst, J.; Hoppe, H.U. A technical and conceptual framework for serious role-playing games in the area of social skill training. Front. Comput. Sci. 2020, 2, p.28. https://doi.org/10.3389/fcomp.2020.00028
  • Chen, H.L.; Wu, C.T. A digital role-playing game for learning: Effects on critical thinking and motivation. Interact. Learn. Environ. 2021, 1–13. https://doi.org/10.1080/10494820.2021.1916765
  • Naivinit, W.; Le Page, C.; Trébuil, G.; Gajaseni, N. Participatory agent-based modeling and simulation of rice production and labor migrations in Northeast Thailand. Environ. Model. Softw 2010, 25(11), 1345–1358. https://doi.org/10.1016/j.envsoft.2010.01.012
  • Barnaud, C.; Le Page, C.; Dumrongrojwatthana, P.; Trébuil, G. Spatial representations are not neutral: Lessons from a par-ticipatory agent-based modelling process in a land-use conflict. Environ. Model. Softw 2013, 45, 150–159. https://doi.org/10.1016/j.envsoft.2011.11.016
  • Putz, F.E.; Sist, P.; Fredericksen, T.; Dykstra, D. Reduced-impact logging: challenges and opportunities. For. Ecol. Manage. 2008, 256(7), 1427–1433. https://doi.org/10.1016/j.foreco.2008.03.036
  • Chen, J.C.; Martin, A.R. Role-play simulations as a transformative methodology in environmental education. J. Transform. Educ. 2015, 13(1), 85–102. https://doi.org/10.1177/1541344614560196
  • Reibelt, L.M.; Waeber, P.O. Approaching human dimensions in lemur conservation at Lake Alaotra, Madagascar. In: Primates. IntechOpen. 2018, 45–66. https://doi.org/10.5772/intechopen.73129
  • García-Barrios, L.; Rivera-Núñez, T.; Cruz-Morales, J.; Urdapilleta-Carrasco, J.; Castro-Salcido, E.; Peña-Azcona, I.; Mar-tínez-López, O.; López-Cruz, A.; Morales, M.; Espinoza, J. The Flow of Peasant Lives: a board game to simulate livelihood strategies and trajectories resulting from complex rural household decisions. Ecol. Soc. 2020, 25(4). https://doi.org/10.5751/ES-11723-250448

We have added the revised version of the manuscript in track change.

Sincerely,

Patrick Waeber,

on behalf of the coauthors

Round 2

Reviewer 1 Report

In order to meet the requirements of a scientific article, the article submitted for review should be supplemented with:

1. the literature review should include information on the methods and goals of testing computer strategy games;

2. the purpose of the research (writing this article) should be supplemented with the categories in which a given strategic game was analyzed. During a comprehensive analysis of games, we test e.g. also sound and graphics, and these elements seem to have not been studied;

3. the methodology for which forests/culture of forest management this game is intended for.

4. in the methodology, specify what elements of the game were tested;

5. in the methodology, describe thefeed back of the game by students - they told their own impressions of the game, was a targeted interview or test conducted?; indiwidual or in grup?;

6. give a brief description of the testers. We know that students, but what basics of forestry they already had. Are they graduating or starting out?

The upper part of the figures 1. should also be corrected. It is illegible.

Author Response

Rebuttal Round 2

We extend our genuine appreciation to the reviewers for their valuable comments, which have significantly strengthened the robustness and clarity of our work. We carefully considered each comment raised by the reviewers and have provided a thorough response addressing them below.

Reviewer 1.

In order to meet the requirements of a scientific article, the article submitted for review should be supplemented with:

  1. the literature review should include information on the methods and goals of testing computer strategy games;

Authors: We understand your suggestion to include information on testing computer strategy games in the literature review. However, we would like to clarify that MineSet is a tabletop role-playing game, not a computer game. We conducted online sessions because the students were unable to attend the university in person, necessitating remote communication and collaboration. We have clarified this in the text.

  1. the purpose of the research (writing this article) should be supplemented with the categories in which a given strategic game was analyzed. During a comprehensive analysis of games, we test e.g. also sound and graphics, and these elements seem to have not been studied;

Authors: We state in the manuscript: “Our focus is on documenting and exploring the game's potential as an educational tool within the classroom setting. By immersing graduate students in a virtual reality of forest management, the game effectively guides them through evolving challenges, facilitating the development of decision-making skills relevant to real-world forest management scenarios. We provide a detailed account of the game's development, application, practical implementation, and the potential benefits it offers for forest education.” The game itself was not tested in this study. But we have written the following in the manuscript “In this study, we employed a customized role-play game called MineSet, which specifically addresses the dynamics of central African forests and aims to educate participants about the underlying causes of deforestation and other global challenges [24,25]. Developed through a participatory modeling approach known as Companion Modeling [26], this constructivist-based game (cf. [61]) is not an off-the-shelf game but rather tailored and created to address a specific and intricate resource management issue in the Congo Basin. The MineSet game development, verification, and validation process included the active participation of various stakeholders from the region. Their involvement ensured that the game's assumptions were grounded and relevant actors and resources and interactions properly calibrated.

Our focus is on assessing the students' understanding of their actions, decisions, and the consequences through extended debriefing sessions. We have emphasized the importance of debriefings in the manuscript.

  1. the methodology for which forests/culture of forest management this game is intended for.

Authors: In MineSet, we use the term 'forests' without explicitly specifying their specific types. However, we do make a distinction between High density forests and Low density forests, as described in the Methodology section and in Figure caption 3. This simplification allows us to apply the game in diverse regions beyond the Congo Basin while still maintaining its meaningfulness. The real value lies in the ensuing discussions that emerge during gameplay. These discussions provide an opportunity to delve deeper into the complexities of diversity, environment, bioclimates, forest types, and other pertinent factors. We have highlighted and addressed this aspect multiple times throughout the manuscript.

  1. in the methodology, specify what elements of the game were tested;

Authors: We have added this to Methodology: “Our approach involves allowing participants to experience the game firsthand, focusing on their individual experiences. The pivotal aspect of the educational process lies in the subsequent debriefing sessions, where in-depth discussions take place. As stated by [55], "learning begins after the game ends." These debriefing sessions serve as the critical moment for reflection and learning. The debriefings occur after each game round and typically extend over several hours. These debriefings involve group discussions that are facilitated by the teachers. To enhance the discussions, an online whiteboard tool called Mural is employed. It serves as a collaborative platform where participants can share their impressions, experiences, and engage in exercises that connect the gameplay to evidence-based and scientific articles. The use of Mural aids in documenting and visually capturing the key points discussed during the debriefings.” We also elaborate in the subsection Limitations (5.3) why it is challenging to have predefined assessment grids when using games like MineSet. We further write “This particular game setup poses challenges when it comes to assessing the impact of the game on educational outcomes. Therefore, it is crucial to emphasize that when games like MineSet are employed in a teaching context, they require careful facilitation and extended debriefing periods, which is at least as important as the gaming itself [55]. These debriefing sessions serve the purpose of establishing meaningful connections between the events and experiences that occurred within the game, validated educational material, and published research outcomes. By engaging in these comprehensive debriefing discussions, educators can better assess the educational impact of MineSet, ensuring that it is leveraged to its full potential for effective learning.

  1. in the methodology, describe thefeed back of the game by students - they told their own impressions of the game, was a targeted interview or test conducted?; indiwidual or in grup?;

Authors: The feedback was collected through group discussions, where students shared their impressions and experiences of the game. We have clarified this in the text.

  1. give a brief description of the testers. We know that students, but what basics of forestry they already had. Are they graduating or starting out?

Authors: The testers were master's level graduate students in their final year of study. They already possess a solid foundation in forestry. We have clarified this in the text.

Reviewer 2 Report

The text has been significantly revised towards a classic article, the authors have taken great care in addressing my comments, for which I thank them very much. I have included detailed comments on the text below. The line numbering refers to the text with the authors' visible changes, not the final version!

Introduction

L49 - maybe write "Professional forest education" to emphasise that this is about educating the people who will manage the forests in the future. Once again, I stress that in some countries forest education for the public (not as professional as foresters, of course) is also widely implemented

L68-69, L90-91 and onwards - unexplained abbreviations are still used (immediately after the use of the full name in brackets its abbreviation should be given, after that the abbreviations themselves can be used). Please correct this everywhere

L77 - 'forest education' - please add 'of graduate students' so that there is no doubt. In Poland, for example, forest education of the public is very much developed, and this is not the same as education of professionals, as the authors of the article write about

Literature Review

L162-163 – revisiting or revising?

L167-170 - Was the game developed in some collaboration with people from the region described, or is it purely theoretical (assumptions prepared by authors from outside the region)?

Materials and Methods

L202-204 - if there are 5-14 players and 5-7 logging companies, that means one logging company can have 2 CEOs? how do they work together? and where did the mining companies disappear to? they are ultimately gone? It is also unclear how to link the information about 5-7 logging companies with the further information about "9 concessions, which fall under the jurisdiction of the Ministry of Forestry" (L227-228)? The authors only explained this in their response to the review, please add in the text: "a logging company can hold more than one concession". Well, and there is no information there (L227-232) about mining concessions - why, if they are in Fig. 1? please clarify these ambiguities further in the text

L233-235 - it has to be with the consent of the concession owner for such a change, or can it be a top-down decision even without the owner's consent?

L240 - "others" means the other owners of the concession? perhaps worth adding to make it precise

L242-243 – after this sentence it is worth adding at least part of the explanation from the review response: „In the game, the students assumed the roles of CEOs and had the authority to make important decisions, including when and where to construct roads and which timber to harvest. However, there was a specific rule that the students were not aware of beforehand: if they opened up forests for logging operations, one of the consequences was the arrival of migrants, represented as tokens in the game. This consequence became evident to the players during gameplay, introducing an element of surprise.” This is important because it explains in more detail the mechanism of how this game works.

L278 - "mining and logging companies" or "mining or logging companies"? can one player have these different companies at the same time?

L307-310 - these are two sentences about the same thing, please merge them somehow

L316-317 - are there any consequences provided for if a concession holder drives a species to extinction? Can these species not be reintroduced?

L318 - this subtitle (subsection) looks artificial, I think it can be dropped or woven into the first sentence of a further paragraph

L332 - it should be added at the end what post-game conversations/discussions the tutors had with the students, that the students' impressions and opinions were recorded and used in the rest of the article - so that it is all coherent in the article and so that there is no impression that the fact that the students played the game is of little relevance to the substance of the article, because it is mainly based on a literature review anyway. Since the authors qualified it like an article and not a literature review, it must have that creative/research element. The authors' text can be used: „Our approach involved allowing participants to experience the game firsthand, focusing on their individual experiences. The pivotal aspect of the educational process lies in the subsequent debriefing sessions, where in-depth discussions take place. As stated by Garcia et al. (2016), "learning begins after the game ends." These debriefing sessions serve as the critical moment for reflection and learning.” And at the end add more about what happened at these debriefing sessions, as I wrote at the beginning of this comment. This needs to be included in the description of the methodology.

Results

L350 (and L388) - 'or land' - no such category in Fig. 3, but Urbanized areas, Shallow waters and Deep waters

L351 - this source should be cited as per journal requirements

L339 - a comment should be added here (after the first sentence) from the authors' response to the previous review, explaining where the greener parts on the map might be coming from: "A smaller number of players would result in slower change within the game. Concessions fall under the jurisdiction of the Ministry of Forestry. If a concession is not utilised, it is likely to still contain high-density forests, although this depends on whether roads pass through the area or not."

Figs. 4, 5, 6 - are more blurred than in the previous version of the article, please restore their previous quality. Fig. 6 - why is there no information on it for 2010 and 2020?

L456 - the source of the information should be given as a numerical citation

L475 - it is worth adding at the end of the paragraph (then it will be coherent) that the activities mentioned can be positively perceived by potential tourists at the same time. And would it be possible to include (in game) fees from tourists for the opportunity to visit such more natural forests?

Discussion

L620-635 - I would move this right after L569. It is currently the case that attention is first drawn to certain details of the game, and only then is the justification found that this game in general is characterised by "an effective transformative environmental education approach". Rather, it should be "from general to specific"

L658 (or elsewhere in subsection 5.2) - it is worth adding information from the review response: „While the students in our presented gameplay of MineSet remained in their roles as logging and mining companies, we did engage in discussions where we explored hypothetical scenarios and different perspectives. These discussions raised the possibility of role swaps and the implications they might have for real stakeholders, including the potential for participants to take on roles that differ or even oppose their usual real-life roles.” Somehow I can't quickly find this in the Discussion, and it's important because it explains "the possible role swaps" from subsection 5.4

L676-677 - not a very clear sentence - what specific risks? what previous discussion?

L757-786 - this is a theory with too few references to the educational process of these six students, please add more such references, as the authors did in Subsection 5.2

L794 - you might want to add after "international forest governance" an explanation that this refers to internationally accepted solutions related to the CBD, forest certification. Otherwise the reader might understand it as, for example, the joint management by countries of forests from their territories

Author Response

Rebuttal Round 2

We extend our genuine appreciation to the reviewers for their valuable comments, which have significantly strengthened the robustness and clarity of our work. We carefully considered each comment raised by the reviewers and have provided a thorough response addressing them below.

Reviewer 2.

The text has been significantly revised towards a classic article, the authors have taken great care in addressing my comments, for which I thank them very much. I have included detailed comments on the text below. The line numbering refers to the text with the authors' visible changes, not the final version!

Introduction

L49 - maybe write "Professional forest education" to emphasise that this is about educating the people who will manage the forests in the future. Once again, I stress that in some countries forest education for the public (not as professional as foresters, of course) is also widely implemented

Authors: done

L68-69, L90-91 and onwards - unexplained abbreviations are still used (immediately after the use of the full name in brackets its abbreviation should be given, after that the abbreviations themselves can be used). Please correct this everywhere

Authors: done

L77 - 'forest education' - please add 'of graduate students' so that there is no doubt. In Poland, for example, forest education of the public is very much developed, and this is not the same as education of professionals, as the authors of the article write about

Authors: done

Literature Review

L162-163 – revisiting or revising?

Authors: "Revisiting" is the intended term here; it emphasizes that students (players in general) have the opportunity to reconsider and reevaluate their assumptions during the gameplay. It signifies a process of rethinking what they believe to be true, while also acknowledging that their experiences within the game may prompt adjustments to those beliefs. In other words, the term "revisiting" highlights the dynamic nature of the learning process and the potential for changes in understanding based on the gameplay experiences. We have modified text, which reads now “The integration of RPGs in education offers the opportunity to simulate group dynamics and present unique scenarios that foster creativity and critical thinking (e.g., encouraging players to revisit their assumptions and make adjustments and modifications to their existing beliefs as a result) [55].

L167-170 - Was the game developed in some collaboration with people from the region described, or is it purely theoretical (assumptions prepared by authors from outside the region)?

Authors: We have clarified in text “The MineSet game development, verification, and validation process included the active participation of various stakeholders from the region. Their involvement ensured that the game's assumptions were grounded and properly calibrated.

Materials and Methods

L202-204 - if there are 5-14 players and 5-7 logging companies, that means one logging company can have 2 CEOs? how do they work together? and where did the mining companies disappear to? they are ultimately gone? It is also unclear how to link the information about 5-7 logging companies with the further information about "9 concessions, which fall under the jurisdiction of the Ministry of Forestry" (L227-228)? The authors only explained this in their response to the review, please add in the text: "a logging company can hold more than one concession". Well, and there is no information there (L227-232) about mining concessions - why, if they are in Fig. 1? please clarify these ambiguities further in the text

Authors: We have clarified this in the text: “The game allows for 5 to 14 players, with players assuming the role of a Chief Executive Officer (CEO) representing a logging or mining company. There are a total of 5 to 7 logging companies available, and if multiple players choose the same company, they act as a single CEO. Additionally, logging companies have the option to venture into mining operations. The logging aspect of the game involves interactions with the Ministry of Forests, while mining activities require permits and rights obtained from the Ministry of Mining. A company can hold more than one concession.” We have further modified this sentence, which reads: “In total there are 9 concessions, which fall under the jurisdiction of the Ministry of Forestry; mining concessions are overlapping and fall under the jurisdiction of the Ministry of Mining.

L233-235 - it has to be with the consent of the concession owner for such a change, or can it be a top-down decision even without the owner's consent?

Authors: We have clarified within text: “In the game, logging concessions can be allocated to companies through leasing agreements. These leasing agreements determine which companies have the rights to operate in specific areas. However, there are also concessions that are not leased to any company. In the gameplay dynamics, players have the opportunity to engage in discussions and negotiations with the Ministry of Forestry regarding the transformation of certain concessions into protected areas. This introduces concepts such as land sparing, leakage, and strict conservation into the game.

L240 - "others" means the other owners of the concession? perhaps worth adding to make it precise

Authors: We have replaced “others” with “amongst each other”

L242-243 – after this sentence it is worth adding at least part of the explanation from the review response: „In the game, the students assumed the roles of CEOs and had the authority to make important decisions, including when and where to construct roads and which timber to harvest. However, there was a specific rule that the students were not aware of beforehand: if they opened up forests for logging operations, one of the consequences was the arrival of migrants, represented as tokens in the game. This consequence became evident to the players during gameplay, introducing an element of surprise.” This is important because it explains in more detail the mechanism of how this game works.

Authors: We have added this: “For instance, if a CEO chooses to allow logging operations by constructing roads through previously undisturbed forests, one of the outcomes is the arrival of migrants, represented as tokens in the game. This consequence becomes apparent to the players during gameplay, introducing an element of surprise and unpredictability.

L278 - "mining and logging companies" or "mining or logging companies"? can one player have these different companies at the same time?

Authors: A company can do both if it has requested and obtained the permits from respective ministry. We have clarified this in the text.

L307-310 - these are two sentences about the same thing, please merge them somehow

Authors: We have modified it; it reads now “The MineSet model incorporates mechanisms that simulate the processes of deforestation, forest degradation, natural growth, and restoration within the game. With these simplified rules, the model effectively reflects these four essential processes [24].”

L316-317 - are there any consequences provided for if a concession holder drives a species to extinction? Can these species not be reintroduced?

Authors: In the game, the consequences of a species going extinct are not explicitly accounted for. It would be challenging to track the main responsible entity (e.g., a logging company) for the disappearance of a species within the game. However, companies within the game have the option to hire biologists, represented by the teachers, to conduct environmental assessments. This mechanism aims to highlight the importance of biodiversity within the game and make it visible to players who might otherwise solely focus on changes in forest cover. Forests, especially intact forests (high density forests), serve as proxies for biodiversity in the game. The underlying game mechanics also allow forests to regenerate and return to high-density levels given sufficient time. However, it is important to note that habitat suitability alone does not guarantee the presence of a species, and if a species is extirpated (locally extinct), its return is not automatically assured.

The possibility of reintroduction as a conservation strategy could be considered for future additions to the game. During gameplay, discussions or considerations related to reintroduction could be initiated if desired by the companies (players). However, it is important to acknowledge that the practical implementation of reintroduction strategies within the game may present challenges. While reintroduction was not brought up by players during the game sessions discussed in this paper, it remains a potential topic for exploration and discussion within the game's framework and during debriefings.

L318 - this subtitle (subsection) looks artificial, I think it can be dropped or woven into the first sentence of a further paragraph

Authors: We agree. We have deleted the title.

L332 - it should be added at the end what post-game conversations/discussions the tutors had with the students, that the students' impressions and opinions were recorded and used in the rest of the article - so that it is all coherent in the article and so that there is no impression that the fact that the students played the game is of little relevance to the substance of the article, because it is mainly based on a literature review anyway. Since the authors qualified it like an article and not a literature review, it must have that creative/research element. The authors' text can be used: „Our approach involved allowing participants to experience the game firsthand, focusing on their individual experiences. The pivotal aspect of the educational process lies in the subsequent debriefing sessions, where in-depth discussions take place. As stated by Garcia et al. (2016), "learning begins after the game ends." These debriefing sessions serve as the critical moment for reflection and learning.” And at the end add more about what happened at these debriefing sessions, as I wrote at the beginning of this comment. This needs to be included in the description of the methodology.

Authors: we have added “Our approach involves allowing participants to experience the game firsthand, focusing on their individual experiences. The pivotal aspect of the educational process lies in the subsequent debriefing sessions, where in-depth discussions take place. As stated by [55], "learning begins after the game ends." These debriefing sessions serve as the critical moment for reflection and learning. The debriefings occur after each game round and typically extend over several hours. These debriefings involve group discussions that are facilitated by the teachers. To enhance the discussions, an online whiteboard tool called Mural is employed. It serves as a collaborative platform where participants can share their impressions, experiences, and engage in exercises that connect the gameplay to evidence-based and scientific articles. The use of Mural aids in documenting and visually capturing the key points discussed during the debriefings.

Results

L350 (and L388) - 'or land' - no such category in Fig. 3, but Urbanized areas, Shallow waters and Deep waters

Authors: There are six landtypes used in the game and described in the legend. The terminology and definition of forested landtypes is based on the High Carbon Stock Approach.

L351 - this source should be cited as per journal requirements

Authors: done

L339 - a comment should be added here (after the first sentence) from the authors' response to the previous review, explaining where the greener parts on the map might be coming from: "A smaller number of players would result in slower change within the game. Concessions fall under the jurisdiction of the Ministry of Forestry. If a concession is not utilised, it is likely to still contain high-density forests, although this depends on whether roads pass through the area or not."

Authors: We have added this to the caption Fig.3: “If a concession, composed of seven hexagons (boundaries not shown here), remains unutilized in the game, it is likely to still consist of High-density forests (dark green). However, the presence or absence of roads passing through the area also impacts the actual condition of the forests.

Figs. 4, 5, 6 - are more blurred than in the previous version of the article, please restore their previous quality. Fig. 6 - why is there no information on it for 2010 and 2020?

Authors: We apologize for the decreased quality of the figures in the current version of the article. We have replaced the figures with new ones and quality should be good now.

Regarding Figure 6, we collected data up to 2020, but regrettably, the information for the years 2010 and 2020 is missing. While this may be seen as a limitation, we believe that the figure still serves its purpose as a visually appealing illustration of game outcomes. It effectively presents the trends and patterns within the available data. We understand that the missing years may raise questions, but we consider the figure valuable for conveying key information and insights.

L456 - the source of the information should be given as a numerical citation

Authors: done

L475 - it is worth adding at the end of the paragraph (then it will be coherent) that the activities mentioned can be positively perceived by potential tourists at the same time. And would it be possible to include (in game) fees from tourists for the opportunity to visit such more natural forests?

Authors: In the development process of MineSet, we followed a participatory stakeholder approach, involving stakeholders in conceptualizing and gamifying the model, as well as incorporating feedback from beta-testers. The feedback received indicated that the game felt realistic to the participants. Notably, tourism was not included as an element in the game, as depicted in Figure 1, and therefore, it was not part of the game sessions with the students. However, the flexibility of the game allows for the introduction of new elements during gameplay, such as tourism fees. If players express a desire to explore the inclusion of tourism-related aspects, it can be a topic for discussion during the debriefing sessions. For instance, a discussion on whether tourism, particularly ecotourism or nature-based tourism, supports conservation efforts could be initiated. To enrich this dialogue, relevant case studies, such as the one mentioned (e.g., https://journals.plos.org/plosone/article?id=10.1371/journal.pone.0278591), can be linked to provide evidence-based examples and perspectives.

Discussion

L620-635 - I would move this right after L569. It is currently the case that attention is first drawn to certain details of the game, and only then is the justification found that this game in general is characterised by "an effective transformative environmental education approach". Rather, it should be "from general to specific"

Authors: done

L658 (or elsewhere in subsection 5.2) - it is worth adding information from the review response: „While the students in our presented gameplay of MineSet remained in their roles as logging and mining companies, we did engage in discussions where we explored hypothetical scenarios and different perspectives. These discussions raised the possibility of role swaps and the implications they might have for real stakeholders, including the potential for participants to take on roles that differ or even oppose their usual real-life roles.” Somehow I can't quickly find this in the Discussion, and it's important because it explains "the possible role swaps" from subsection 5.4

Authors: We have added this “While the students participating in our MineSet gameplay remained in their designated roles as logging and mining companies, we also engaged in discussions where we explored hypothetical scenarios and considered different perspectives. These discussions raised the possibility of role swaps and the implications they might have for real stakeholders, including the potential for participants to take on roles that differ or even oppose their usual real-life roles.

L676-677 - not a very clear sentence - what specific risks? what previous discussion?

Authors: Sentence reads now “When organizing a game workshop centered around games like MineSet, it is crucial to acknowledge the inherent risk that players may learn strategies that primarily benefit themselves but have negative consequences for the environment or other stakeholders. Garcia et al. [23] highlighted these risks by emphasizing the connection to power dynamics and asymmetries and conflicts.

L757-786 - this is a theory with too few references to the educational process of these six students, please add more such references, as the authors did in Subsection 5.2

Authors: We have cited already existing references, and we have added a new one: Garcia, C.A., Savilaakso, S., Verburg, R.W., et al. 2020. The global forest transition as a human affair. One Earth, 2(5), pp.417-428. https://doi.org/10.1016/j.oneear.2020.05.002

L794 - you might want to add after "international forest governance" an explanation that this refers to internationally accepted solutions related to the CBD, forest certification. Otherwise the reader might understand it as, for example, the joint management by countries of forests from their territories

Authors: We have adapted text: “While we have provided an example specifically related to forests, it's important to note that the application of such games can be extended to any complex subject within the realms of environmental management and governance. This includes internationally accepted solutions pertaining to various aspects such as the Convention on Biological Diversity (CBD), forest certification, and more. These games serve as valuable tools for exploring and understanding the intricacies of different environmental issues and their associated management and governance challenges.

Please find revised manuscript in attachment.

Reviewer 4 Report

The author(s) revised the article as suggested.

Author Response

Dear Reviewer,

We sincerely appreciate your thorough feedback provided in round 1. We have noticed that you did not include any comments in review 2, and consequently, we do not have any specific points listed here. However, we would like to take this opportunity to share with you the latest and revised version of our manuscript.

Thank you for your time and consideration.
